# EQUIVARIANT QUANTUM GRAPH NEURAL NETWORK FOR MIXED-INTEGER LINEAR PROGRAMMING

## ABSTRACT

Mixed-integer linear programming (MILP) is an essential task for operation research, especially for combinatorial optimization problems. Apart from the classic non-learning solvers that often resort to heuristics, recent machine learning-based models have been actively studied, and graph neural networks (GNNs) have been dominantly adopted. However, recent literature has shown that the GNNs based on message passing mechanism suffer fundamental expressiveness limitations in MILP instance representation, in the sense that two different MILP instances could be eventually embedded into exactly the same feature. In this paper, we resort to the quantum mechanism and develop a tailored quantum counterpart of GNNs, called equivariant quantum GNN (EQGNN), which can overcome the fundamental limitation of traditional GNNs, i.e., it can distinguish two MILPs that cannot be distinguished by GNNs. Specifically, EQGNN is designed to be the structure of permutation equivariance, which is key to learning the graph-structure data because the solution of an MILP should be reordered consistently with the permutation on the variables. While maintaining equivariance, EQGNN presents a multi-qubit encoding mechanism for encoding features and a parameter-sharing mechanism for graph information interaction. To enhance the expressivity power of the model, EQGNN also introduces an auxiliary layer with an optional number of auxiliary qubits. Experimental results demonstrate the effectiveness of the method in solving MILP problems and the trainability of the model with increasing system scale. Compared with traditional GNNs, EQGNN can achieve better separation power and generalization performance with fewer parameters. The source code will be made publicly available.

## 1 INTRODUCTION

Quantum machine learning (QML) emerges as a promising field which harnesses the principles of quantum mechanics and the power of machine learning (Biamonte et al., 2017; Cerezo et al., 2022). In particular, quantum neural networks (QNN) (Abbas et al., 2021) can be embodied as parameterized quantum circuits (PQC) (Benedetti et al., 2019) executed on current Noisy Intermediate-Scale Quantum (NISQ) devices (Bharti et al., 2022) in a variational training manner using classical (e.g. gradient-based) optimizers. It is still an important and active field to explore the potential advantage of QNNs over their classical counterparts given training data (Yu et al., 2022).

In general, QNNs can be categorized into problem-agnostic and problem-inspired architectures. Problem-agnostic ansatzes, e.g. hardware-efficient ansatz (Kandala et al., 2017), do not depend on problem information and thus usually need strong expressibility (Du et al., 2020). Meanwhile, they are also often more likely to exhibit trainability issues (Holmes et al., 2022), e.g. the barren plateau phenomenon (McClean et al., 2018). In contrast, problem-inspired ansatzes can be designed by prior about the data and problem, which can confine the design space. In particular, to address graph-structured data as commonly encountered in real-world problems, e.g. molecular property prediction (Ryu et al., 2023) and combinatorial optimization (Ye et al., 2023), a number of QNNs (Verdon et al., 2019; Zheng et al., 2021; Ai et al., 2022; Mernyei et al., 2022) have been proposed.

To enable quantum graph learning, Verdon et al. (2019) introduce a class of quantum GNNs using Hamiltonian based on graph structure, while the model is constrained by the specific form of the Hamiltonian. Zheng et al. (2021) and Ai et al. (2022) propose specific quantum circuits and consider the encoding of the high-dimensional features, but neither of their networks ensures the permutation

invariance w.r.t input nodes, which is a vital property for graph tasks. In terms of geometric quantum machine learning (GQML), Schatzki et al. (2022) show incorporating geometric priors via $S_n$-equivariance into QML can heavily restrict the model's search space, mitigating the barren plateau issue and can generalize well with small data. Recently, a theoretical recipe for building equivariant and invariant quantum graph circuits is given by Mernyei et al. (2022), but they do not provide the specific circuit implementation nor consider the edge features of graph in their model design. We leave a detailed discussion on related works in Appendix A including a Table 3 as summary. In fact, there remains an unexplored direction in designing explicitly equivariant and invariant quantum networks for tackling graph problems with multi-dimensional node features and edge weights.

In this paper, we consider a general form that can cover various combinatorial optimization (CO) problems, i.e. mixed-integer linear programming (MILP). The instances can be represented by weighted bipartite graphs with node features (Gasse et al., 2019; Chen et al., 2023a) (see more details in the later preliminary section). Due to the nice properties of permutation invariance of GNNs, it is considered a suitable backbone in various stages of MILP solving processes (Khalil et al., 2022; Gupta et al., 2022; Wang et al., 2023). However, the recent work (Chen et al., 2023b) shows a fundamental limitation in using GNNs to express arbitrary MILP instances: there exists a set of feasible and infeasible MILP instances treated as identical by the GNNs, rendering the GNN incapable of distinguishing their feasibility, as shown in Fig. 2. They called these MILP instances that cannot be distinguished by GNNs as *foldable* MILPs, while MILPs that can be distinguished by GNNs are named as *unfoldable* MILPs. To predict MILP feasibility, optimal values and solutions, GNNs can only restrict the MILP instances to be unfoldable or add random features for foldable MILPs. However, discerning foldable and unfoldable MILPs inherently requires the extra preprocessing techniques.

To this end, we propose a so-called Equivariant Quantum Graph Neural Network (EQGNN) to overcome the fundamental limitation of traditional GNNs, *i.e.*, GNNs based on the message-passing mechanism cannot distinguish pairs of foldable MILP instances. Around 1/4 of the problems in MIPLIB 2017 (Gleixner et al., 2021) involve foldable MILPs. It means that practitioners using GNNs cannot benefit from that if there are foldable MILPs in the dataset of interest (Chen et al., 2023b). In contrast, EQGNN can distinguish graphs that cannot be distinguished by GNNs, such that it is capable of representing general MILP instances. Moreover, EQGNN can be regarded as a QML model that introduces strong relational inductive bias by designing a symmetry-preserving ansatz, and it can used for learning any graph-structure data. **The contributions of this work are:**

**1)** We propose a novel equivariant quantum GNN, which consists of the feature encoding layer, graph message interaction layer, and optional auxiliary layer. The permutation equivariance of the model is key to learning graph-structure data, *e.g.*, the predicted solution of an MILP should be reordered consistently with the permutation on the variables. To ensure the permutation equivariance, EQGNN designs a parameter-sharing mechanism and carefully chooses parametric gates for learning graph information interaction. To encode edge and node features, EQGNN presents a multi-qubit encoding mechanism and repeated encoding mechanism. Moreover, we introduce the auxiliary layer to enhance the expressive power of EQGNN. Experiments show the good trainability of our EQGNN with increasing system scale.

**2)** We show the separation power of EQGNN can surpass that of GNNs in terms of representing MILP graphs. EQGNN can distinguish MILP graphs that cannot be recognized by GNNs using the unique properties of quantum circuits, thereby accurately predicting the feasibility of the general MILPs. Moreover, extensive numerical experiments have shown that our EQGNN achieves faster convergence, utilizes fewer parameters, and attains better generalization with less data compared to GNNs. Based on this, EQGNN holds the potential to advance the field of leveraging quantum computing to assist in the classical methods for MILP solving.

## 2 PRELIMINARIES

**MILP as a weighted bipartite graph with node features.** A general MILP problem can be defined as follows, where $A \in \mathbb{R}^{p \times q}$, $b \in \mathbb{R}^p$, $c \in \mathbb{R}^q$:

$$\min_{x \in \mathbb{R}^q} \ c^\top x, \quad \text{s.t.} \ Ax \circ b, \ l \leq x \leq u, \ x_i \in \mathbb{Z}, \ \forall i \in I, \tag{1}$$

where $l$ and $u$ represent the upper and lower bounds on variables, where $l \in (\mathbb{R} \cup \{-\infty\})^q$, $u \in (\mathbb{R} \cup \{+\infty\})^q$ and $\circ \in \{\leq, =, \geq\}^p$. $I \subseteq \{1, \cdots, q\}$ represents the index set of integer variables.

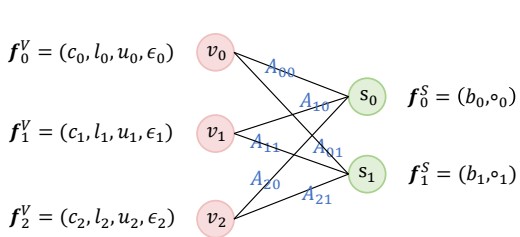
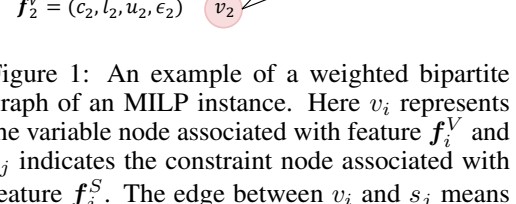
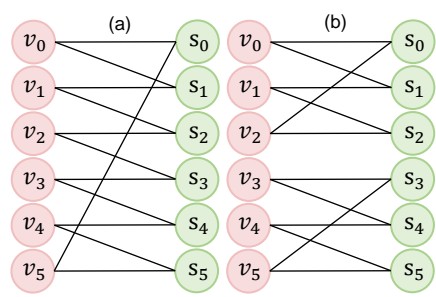

Figure 1: An example of a weighted bipartite graph of an MILP instance. Here $v_i$ represents the variable node associated with feature $\boldsymbol{f}_i^V$ and $s_j$ indicates the constraint node associated with feature $\boldsymbol{f}_j^S$. The edge between $v_i$ and $s_j$ means that the $j$-th constraint involves the $i$-th variable.

Figure 2: An example of foldable MILP instances that cannot be distinguished by the 1-WL test and GNNs. The only difference between these two graphs lies in the connectivity of the edges, resulting in (a) being feasible and (b) being infeasible.

The *feasible solution* is defined as the set $X_{fea} = \{x \in \mathbb{R}^q \mid Ax \circ b, \, l \le x \le u, \, x_i \in \mathbb{Z}, \, \forall i \in I\}$, while $X_{fea} = \emptyset$ means the MILP problem is *infeasible*. Feasible MILPs has *optimal objective value* $y_{obj} = \inf\{c^\top x \mid x \in X_{fea}\}$. If there exists $\hat{x}$ so that $c^\top \hat{x} \le c^\top x, \, \forall x \in X_{fea}$, then $\hat{x}$ is an *optimal solution*. Following the protocol in (Gasse et al., 2019) and (Chen et al., 2023b), we formulate MILP as a *weighted bipartite graph* to interpret variable-constraint relationships, as illustrated in Fig. 1. The vertex set of such graph is $V \cup S$, where $V = \{v_0, \cdots, v_i, \cdots, v_{q-1}\}$ with $v_i$ representing the $i$-th variable and $S = \{s_0, \cdots, s_j, \cdots, s_{p-1}\}$ with $s_j$ representing the $j$-th constraint. The edge connected $v_i$ and $s_j$ has weight $A_{i,j}$. Based on the Eq. (1), the vertex $v_i \in V$ is associated with a feature vector $\boldsymbol{f}_i^V = (c_i, l_i, u_i, \epsilon_i)$, where $\epsilon_i$ represents whether variable $v_i$ takes integer value. The vertex $s_j$ is equipped with a two-dimension vector $\boldsymbol{f}_j^S = (b_j, \circ_j)$. There is no edge between vertices in the same vertex set ($V$ or $S$). The weighted bipartite graph with node features is named as an *MILP-induced graph* or *MILP graph*.

**Foldable MILP instances.** (Chen et al., 2023b) categorizes MILP instances into *foldable* if GNNs cannot distinguish them (*i.e.*, GNN learns two different MILP instances to be the same representation), and the rest of MILPs that can be distinguished by GNNs as *unfoldable* MILP instances. Fig. 1 shows an example of a pair of MILPs in the *foldable* dataset. Assume that $\boldsymbol{f}_i^V = (1, 0, 1, 1)$, for all $v_i \in V$ and $\boldsymbol{f}_j^S = (1, =)$, for all $s_j \in S$ and all edge weights are equal to 1, which means that the only difference between the two bipartite graphs lies in the connectivity of edges. However, these two MILP instances have different feasibility. Fig. 2 (a) is feasible such as $x = (0, 1, 0, 1, 0, 1)$ is a feasible solution, while Fig. 2 (b) is infeasible because there are no integer decision variables that can satisfy the equality constraints $2(x_0 + x_1 + x_2) = 3$. Moreover, these pair of graphs cannot be distinguished by 1-dimensional Weisfeiler-Lehman (1-WL) algorithm (Weisfeiler & Leman, 1968), because each node has two neighbors with the same features and all edge weights also are equal. Since the expressive power of GNNs based on message passing mechanism is upper bounded by the 1-WL algorithm for graph isomorphism testing (Xu et al., 2018; Morris et al., 2019), these pair of 1-WL indistinguishable graphs will cause the GNNs to learn the same representations and yielding the same prediction. Thus directly applying GNNs to represent MILPs may fail on general datasets. See Appendix B for proof.

## 3 EQUIVARIANT QUANTUM GRAPH NEURAL NETWORK

### 3.1 APPROACH OVERVIEW

Fig. 3 shows our Equivariant Quantum Graph Neural Network (EQGNN), with the feature encoding layer, graph message interaction layer, and auxiliary layer. The three layers form a block, iterated repeatedly in the circuit. All the layers in EQGNN adhere to the principle of equivariance, detailed in Sec. 3.6. We study whether EQGNN can map an MILP to its feasibility, optimal objective value, and optimal solution. Predicting feasibility and optimal objective values is a graph-level problem that maps an MILP graph to a value, and predicting the optimal solution is a node-level problem that maps an MILP to a solution vector. The three tasks can use the same structure of EQGNN, except that the graph-level problem requires a permutation-invariant aggregation for the output of the equivariant model. Fig. 10 shows the properties and invariant and equivariant models for MILP graphs.

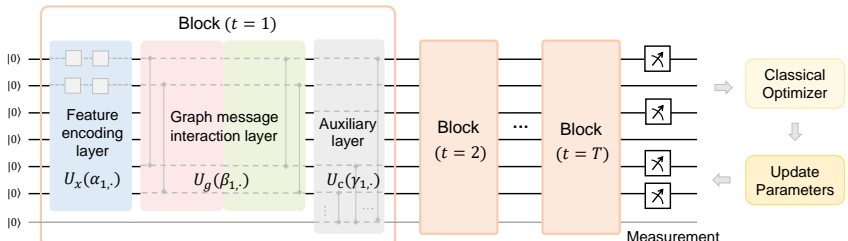

Figure 3: Architecture of our EQGNN for representing MILPs. The feature encoding layer encodes node information into the quantum circuit, and the graph message interaction layer contains variable update layer and constraint update layer. Auxiliary layer is optional and used for enhancing the capacity of the model. All layers are designed to preserve the equivariance of the node permutation.

## 3.2 FEATURE ENCODING LAYER

Recall that an MILP can be encoded into a bipartite graph. The node representing variables has four features $(c, l, u, \epsilon)$, and the constraint node has two features $(b, \circ)$. In our feature encoding layer, we use an angle encoding scheme, which takes the features as the parameters of gates. Moreover, we adopt an alternating encoding involving features and trainable parameters, associating a trainable parameter with each individual feature, thereby harnessing the information of the nodes. We set a multi-qubit encoding mechanism that requires each qubit can only encode at most $\omega$ features. The choice of $\omega$ is directly associated with the circuit width. The smaller the value of $\omega$, the larger the number of qubits required for the quantum circuit, the more expressive the circuit. Fig. 4 shows an example of the feature encoding circuit under the case of two variables ($v_0$ and $v_1$) and two constraints ($s_0$ and $s_1$). $\omega$ is set to 2, so the

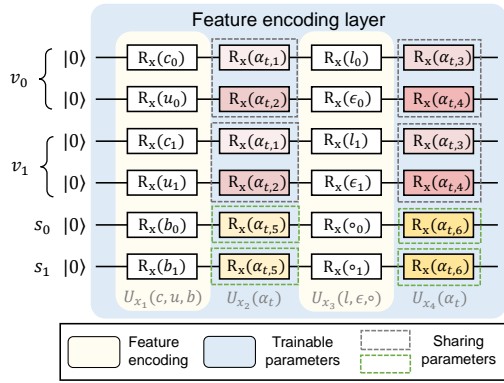

Figure 4: Features are encoded into the circuit by angle encoding, and each feature is associated with a trainable parameter. For permutation equivariance, each feature shares an identical trainable parameter.

features of variables are encoded by two qubits. To ensure the node permutation invariance, variables share one common set of parameters, while constraints share another set of parameters. As shown in Fig. 4, dashed boxes of the same color indicate the same trainable parameters. The unitary matrix of feature encoding layer is denoted as $U_x^t(G, H, \alpha_t) = U_{x_1}(c, u, b) \cdot U_{x_2}(\alpha_t) \cdot U_{x_3}(l, \epsilon, \circ) \cdot U_{x_4}(\alpha_t)$. See the Equation 11 in the Appendix C for their unitary matrix.

## 3.3 QUANTUM GRAPH MESSAGE INTERACTION LAYER

After the feature encoding layer, the quantum graph message interaction layer is used to update the information of qubits representing variables and constraints by acting the two-qubit gate on the qubits connected by an edge. As illustrated in Fig. 5, we show an example of the graph message interaction layer for the MILP graph in the legend (a) in the case of $\omega = 2$. The unitary of the $t$-th graph message interaction layer can be represented by $U_g^t(G, \beta_t) = U_{gv}^t(G, \beta_t) \cdot U_{gs}^t(G, \beta_t)$, where $\beta_t$ is the set of trainable parameters. $U_{gv}^t(G, \beta_t)$ denotes the variable update layer, and $U_{gs}^t(G, \beta_t)$ denotes the constraint update layer. We define

$$U_{g_1}(G, \beta) = \exp(-\mathbf{i}(\sum_{(i,j) \in \mathcal{E}} (A_{i,j} + \beta) Z_{2i} Z_{2q+j})), U_{g_2}(G, \beta) = \exp(-\mathbf{i}(\sum_{(i,j) \in \mathcal{E}} (A_{i,j} + \beta) Z_{2i+1} Z_{2q+j})),$$
$$(2)$$

which indicates that the circuit uses $\mathrm{R}_{ZZ}(\beta)$ gates with learnable parameters to act on two qubits representing two nodes connected by the edge. $U_{g_1}$ denotes the information interaction between the qubit representing the constraint node and the *first* qubit representing the variable node. $U_{g_2}$ denotes the information interaction of the *second* qubit representing the variable node. Then, the variable update layer is defined as

$$U_{gv}^t(G, \beta) = U_{g_1}(G, \beta_{t,1}) \cdot U_{g_2}(G, \beta_{t,2}) \cdot U_{g_3}(\beta_{t,3}), \quad U_{g_3}(\beta_{t,3}) = \bigotimes_{i=0}^{q-1} \mathrm{CR}_{\mathrm{y}}(\beta_{t,3}) \otimes \bigotimes_{j=0}^{p-1} I. \quad (3)$$

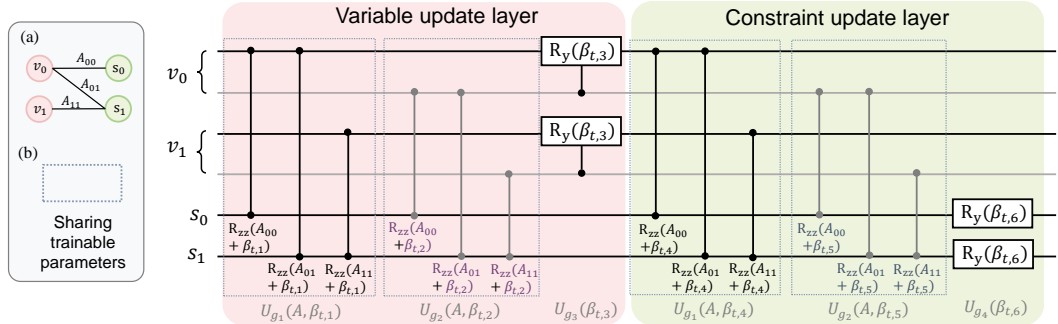

Figure 5: Quantum graph message interaction layer consists of a variable update layer and a constraint update layer. For the bipartite graph in (a) (left top), we present a circuit structure of the layer. The $R_{zz}$ gate acts on the $v_i$ and $s_j$ with $A_{i,j}$, and the trainable parameter $\beta_t$ serving as its rotation parameters. In (b) (left bottom), the dashed box indicates that the gates inside the box share a trainable parameter.

Similarly, the constraint update layer is defined as

$$U_{gs}^t(G, \beta) = U_{g_1}(G, \beta_{t,4}) \cdot U_{g_2}(G, \beta_{t,5}) \cdot U_{g_4}(\beta_{t,6}), \quad U_{g_4}(\beta_{t,6}) = \exp(-\mathbf{i}(\beta_{t,6} Y_{2q+j})). \quad (4)$$

Each sublayer in the $U_g^t$ shares a trainable parameter to maintain the equivariance of our model, which will be further explained in Sec. 3.6.

### 3.4 AUXILIARY LAYER

To further enhance the expressive capacity of the model, we introduce an auxiliary layer, which is optional for different tasks. By adding the auxiliary qubits, we can increase the Hilbert space of the model and further facilitate the interaction of information within the graph. Specifically, each auxiliary qubit is connected to all other nodes through $R_{zz}$ gates. For the two qubits representing variables, trainable parameters $\gamma_{t,1}$ and $\gamma_{t,2}$ are assigned, while parameter $\gamma_{t,3}$ is assigned to the qubit representing constraints. Following the application of the two-qubit gates, single-qubit gates $R_y(\gamma_{t,3})$ and $R_z(\gamma_{t,4})$ are applied to the auxiliary qubits. We can choose a varying number of auxiliary qubits.

### 3.5 MEASUREMENT LAYER AND OPTIMIZATION

The feature encoding layer, graph message interaction layer, and auxiliary layer form a block. After this block is iteratively repeated $T$ times, Pauli-Z measurement is required to act on the prescribed qubits. If the number of qubits representing nodes is more than 1, such as the node $v_1$ in Fig. 5, we will add the control gate at the end of the node update layer, such as the controlled $R_y$ gate in Fig. 5. Then, at the end of the circuit, we only measure the first qubit representing the node. The measurements corresponding to Fig. 5 are shown at the end of the overall architecture diagram in Fig. 3. As we can see, the measurement operation of the model acts on $q + p$ qubits, and we can obtain $q + p$ output values, where $q$ and $p$ are the numbers of decision variable nodes and constraint nodes, respectively.

We can represent the MILP graph as $G = (V \cup S, A)$ and $\mathcal{G}_{q,p}$ as the collection of all such weighted bipartite graphs whose two vertex groups have size $q$ and $p$, respectively. All the vertex features are stacked together as $H = (\boldsymbol{f}_1^V, ..., \boldsymbol{f}_q^V, \boldsymbol{f}_1^S, ..., \boldsymbol{f}_p^S) \in \mathcal{H}_q^V \times \mathcal{H}_p^S$. Thereby, the weighted bipartite graph with vertex features $(G, H) \in \mathcal{G}_{q,p} \times \mathcal{H}_q^V \times \mathcal{H}_p^S$ contains all information in the MILP problem. The proposed model can be described as a mapping $\Phi : \mathcal{G}_{q,p} \times \mathcal{H}_q^V \times \mathcal{H}_p^S \to \mathbb{R}^{q+p}$, i.e., $\Phi(G, H) = \{\langle 0|U_{\boldsymbol{\theta}}^\dagger(G, H)|O_i|U_{\boldsymbol{\theta}}(G, H)|0\rangle\}_{i=0}^{q+p-1}$, where $\boldsymbol{\theta}$ denotes the set of trainable parameters $(\boldsymbol{\alpha}, \boldsymbol{\beta}, \boldsymbol{\gamma})$, and $U_{\boldsymbol{\theta}}(G, H)$ is the unitary matrix of the proposed model. $O_i$ represents $i$-th measurement, e.g., when $\omega$ is equal to 2, $O_0 = Z_0 \otimes I_1 \otimes ... \otimes I_{2q+p-1}$ indicates that Pauli-Z measurement is acted on the qubit representing the first variable, and $O_1 = I_0 \otimes I_1 \otimes Z_2 \otimes ... \otimes I_{2q+p-1}$ for the second variable node, and $O_{q+p-1} = I_0 \otimes I_1 \otimes ... \otimes Z_{2q+p-1}$ for the last constraint node. The output of EQGNN is defined as $\{\Phi(G, H)_i\}_{i=0}^{p+q-1}$. For predicting feasibility, optimal value and optimal solution, we defined $\phi_{sol}(G, H) = \{\Phi(G, H)_i\}_{i=0}^{q-1}$, and $\phi_{fea}(G, H) = \phi_{obj}(G, H) = \sum_{i=0}^{q+p-1} \Phi(G, H)_i$. As we can see, the three tasks use the same structure of EQGNN and the same measurements, but use different ways to utilize the information obtained by measurements.

**For predicting the feasibility,** $\hat{y}_{fea} = \phi_{fea}(G, H)$, we utilize the negative log-likelihood as the loss function to train EQGNN. In the testing, we set an indicator function

$$\mathbb{I}_{\hat{y}_{fea}>1/2} = \begin{cases} 0, & \hat{y}_{fea} \leq 1/2 \\ 1, & \hat{y}_{fea} > 1/2 \end{cases} \tag{5}$$

to calculate rate of errors, *i.e.*, $\frac{1}{M}(\sum_{m=0}^{M-1} y_{fea}^m \cdot \mathbb{I}_{\hat{y}_{fea}>1/2}^m)$, which is used to evaluate the number of correct predictions for feasibility, where $M$ indicates the number of tested MILP instances.

**For predicting the optimal solutions,** $\hat{\boldsymbol{y}}_{sol} = \lambda\phi_{\text{sol}}(G, H)$, where $\lambda$ is the maximum range of variables of training sample, *i.e.*, $\max\{\{\{abs(l_i^n), abs(u_i^n)\}_{i=0}^{q-1}\}_{n=0}^{N-1}\}$. We use the mean square error as the training and testing metric, *i.e.*, $\frac{1}{Mq}\sum_{m=0}^{M-1}\|\boldsymbol{y}_{sol} - \hat{\boldsymbol{y}}_{sol}\|_2^2$, where $\boldsymbol{y}_{sol}$ is the groundtruth.

**For predicting the optimal values,** $\hat{y}_{obj} = \delta\lambda\phi_{\text{obj}}(G, H)$, where $\delta = \max\{\{\{c_i^n\}_{i=0}^{q-1}\}_{n=0}^{N-1}\}$ is the maximum range of coefficients of training sample. We also use the mean square error to train or evaluate, *i.e.*, $\frac{1}{M}\sum_{m=0}^{M-1}(y_{obj}^m - \hat{y}_{obj}^m)^2$.

### 3.6 Equivariance and invariance of the proposed model

**Definition 1.** *Equivariance. The function $\phi$ is permutation equivariant if $\phi(\sigma(G, H)) = \sigma(\phi(G, H))$ for any $\sigma \in \mathcal{S}^n$, where $\mathcal{S}^n$ is the group contains all permutations on the nodes of $G$, and $\sigma(G, H)$ denotes the reordered graph with permutations $\sigma$.*

**Definition 2.** *Invariance. $\phi$ is permutation invariant if $\phi(\sigma(G, H)) = \phi(G, H)$ for any $\sigma \in \mathcal{S}^n$.*

**Theorem 1.** *(Invariance from equivariance) If a function $\phi_1$ is permutation equivariant, there exists a permutation-invariance operation $\phi_2$, such that $\phi_2(\phi_1(G, H))$ is permutation invariant.*

Given $(a_1, a_2, ..., a_n)$ as the output of $\phi_1(G, H)$, and the permutation-invariance operation is summation. $\phi_2(\phi_1(G, H)) = \sum_{i=0}^{n-1} a_i$, and $\phi_1(\phi_2(\sigma(G, H))) = \sum_{i=0}^{n-1} a_{\sigma(i)}$. Since $\sum_{i=0}^{n-1} a_{\sigma(i)} = \sum_{i=0}^{n-1} a_i$, $\phi_2(\phi_1(G, H)) = \phi_2(\phi_1(\sigma(G, H)))$ indicates $\phi_2(\phi_1(G, H))$ is permutation invariant.

**Definition 3.** *A $T$-layered QNN ($U_{\boldsymbol{\theta}} = \prod_{t=0}^{T-1} U_{\boldsymbol{\theta}}^t$) is permutation equivariant iff each layer is permutation equivariant. The layer $U_{\boldsymbol{\theta}}^t$ of a QNN is permutation equivariant iff $U_{\boldsymbol{\theta}}^t(\sigma(G, H)) = R(\sigma)U_{\boldsymbol{\theta}}^t(G, H)$, where $R$ is the unitary representation of $\sigma$ on quantum states.*

It means that we can decompose the QNN into multiple sublayers to prove the equivariance. EQGNN has $T$ identical blocks with respective trainable parameters, and each block consists of three layers. Moreover, each layer has sublayers, such as the feature encoding layer $U_x^t(G, H, \alpha_t) = U_{x_1}(c, u, b) \cdot U_{x_2}(\alpha_t) \cdot U_{x_3}(l, \epsilon, \circ) \cdot U_{x_4}(\alpha_t)$, and the message interaction layer $U_g^t(G, \beta) = U_{g_1}(G, \beta_{t,1}) \cdot U_{g_2}(G, \beta_{t,2}) \cdot U_{g_3}(\beta_{t,3}) \cdot U_{g_1}(G, \beta_{t,4}) \cdot U_{g_2}(G, \beta_{t,5}) \cdot U_{g_4}(\beta_{t,6})$. The whole model conforms to permutation equivariance by ensuring that each layer conforms to equivariance.

**Definition 4.** *There are two types of layers in $U_{\boldsymbol{\theta}}(G, H)$, one is independent of the order of nodes, and the other is related to the order of nodes. We define the layer independent of the node order as $U^t(\boldsymbol{\theta})$ and the layer related to node order as $U^t(G, H, \boldsymbol{\theta})$.*

For different permutation of input nodes, the layer $U^t(\boldsymbol{\theta})$ is identical. In the proposed EQGNN, $U_{x_2}(\alpha_t), U_{x_4}(\alpha_t), U_{g_3}(\beta_{t,3}), U_{g_4}(\beta_{t,6})$ and auxiliary layer are the layers that are independent of the permutation of input, which is implemented by sharing a single trainable parameter over all qubits representing variables or constraints, making the order of variables and constraints unimportant. This shows the importance of the proposed **parameter-sharing mechanism**. Therefore, we only need to prove the equivariance of the layer $U^t(G, H, \boldsymbol{\theta})$ in the proposed EQGNN, including $U_{x_1}(c, u, b), U_{x_3}(l, \epsilon, \circ), U_{g_1}(G, \beta_t), U_{g_2}(G, \beta_t)$.

**Theorem 2.** *$U_{x_1}(c, u, b)$ and $U_{x_3}(l, \epsilon, \circ)$ are equivariant w.r.t. the permutations $\sigma_v \in \mathcal{S}^q$ and $\sigma_s \in \mathcal{S}^p$, where $\mathcal{S}^q$ is defined as the group contains all permutations on the variables of MILP and $\mathcal{S}^p$ is defined as the group contains all permutations on the constraints.*

$U_{x_1}(c, u, b)$ and $U_{x_3}(l, \epsilon, \circ)$ can be regarded as the first and third layers in Figure 4, and we can see Appendix F for the proof.

**Theorem 3.** *$U_{g_1}(G, \beta_t)$ and $U_{g_2}(G, \beta_t)$ are equivariant w.r.t. the permutations $\sigma_v$ and $\sigma_s$.*

*Proof.* The difference with $U_{x1}$ and $U_{x3}$ is that $U_{g1}$ and $U_{g2}$ involve the topology of the graph. $\sigma_v\sigma_s(G) = (\sigma_v(V) \cup \sigma_s(S), A')$, where $A' = P_{\sigma_v}AP_{\sigma_s}^T$, and $P_{\sigma_v}$ and $P_{\sigma_s}$ are the permutation

matrix of $\sigma_v$ and $\sigma_s$, respectively. We can obtain $A'_{\sigma_v(i),\sigma_s(j)} = A_{i,j}$. The original edge $(i,j) \in \mathcal{E}$, and the transformed edges $(\sigma_v(i), \sigma_s(j)) \in \mathcal{E}'$. According to $U_{g_1}(G, \beta_t)$ in Eq. 2,

$$U_{g_1}(\sigma_v\sigma_s(G), \beta_t) = \exp(-\mathbf{i}(\sum_{(\sigma_v(i),\sigma_s(j))\in\mathcal{E}'} (A'_{\sigma_v(i),\sigma_s(j)} + \beta_t)Z_{2\sigma_v(i)}Z_{2q+\sigma_v(j)})). \quad (6)$$

Based on the unitary representation of permutation $R(\sigma_v)$ and $R(\sigma_s)$, we can obtain

$$(R(\sigma_v) \otimes R(\sigma_s))(U_{g_1}(G, \beta_t)) = \exp(-\mathbf{i}(\sum_{(i,j)\in\mathcal{E}} (A_{i,j} + \beta_t)Z_{2\sigma_v(i)}Z_{2q+\sigma_v(j)})). \quad (7)$$

Although $A_{i,j} = A'_{\sigma_v(i),\sigma_s(j)}$, the order of edges in $\mathcal{E}$ and $\mathcal{E}'$ may be different. Therefore, we need to guarantee *the permutation invariance of edges*. For example, it should satisfy

$$\exp(-\mathbf{i}(A_{i_1,j_1}Z_{2i_1}Z_{2q+j_1} + A_{i_2,j_2}Z_{2i_2}Z_{2q+j_2})) = \exp(-\mathbf{i}(A_{i_2,j_2}Z_{2i_2}Z_{2q+j_2} + A_{i_1,j_1}Z_{2i_1}Z_{2q+j_1})). \quad (8)$$

Since $\exp(-\mathbf{i}A_{i,j}Z_iZ_j)$ is diagonal, and all diagonal matrices commute, the equation holds. Thus, Eq. 6 and Eq. 7 can be proven to be equal. Note that $\mathrm{R_{zz}}$ gate is not the only option to preserve the permutation invariant of edges, and the two-qubit gates that can commute in the circuit, such as $\mathrm{R_{yy}}$ and $\mathrm{R_{xx}}$, are able to be used to learn the graph information interaction. In a similar way $U_{g_2}(G, \beta_t)$ can also be proven to be permutation equivariant w.r.t. $\sigma_v$ and $\sigma_s$. $\square$

By Theorem 2 and 3, we can obtain the layers related to the input order are permutation equivariance. Then, by Definition 3 and 4, we can obtain the permutation equivariance of our EQGNN.

## 4 EXPERIMENTS

We first compare the separation power and expressive power of our EQGNN and the GNN used in (Chen et al., 2023b) on the foldable and unfoldable MILP datasets, respectively. Then, the performances of different schemes of quantum neural network are compared on the MILP tasks. We also conduct an ablation study for EQGNN and analyze the trainability of EQGNN. All the experiments are performed on a single machine with 4 physical CPUs with 224 cores Intel(R) Xeon(R) Platinum 8276 CPU @ 2.20GHz, and a GPU (NVIDIA A100). Source code is written using TorchQauntum (Wang et al., 2022a), which is a Pytorch-based library for quantum computing.

### 4.1 EXPERIMENTAL DETAIL

A classical optimizer Adam (Kingma & Ba, 2014) with an initial learning rate of $0.1$ is used to find the optimal parameters of quantum circuits, including $\alpha, \beta$, and $\gamma$ and batch size is set at $32$. The proposed model has a hyperparameter to control the number of parameters, i.e., the number of blocks $T$. The number of our parameters in predicting feasibility is $12T$. The GNN (Chen et al., 2023b) also has one hyperparameter that controls the number of parameters, i.e., the embedding size $d$. Take predicting feasibility as an example, the number of their parameters is $30d^2 + 30d$. Therefore, we vary these two hyperparameters separately to compare their performance results. In all our experiments, we first gradually increase the embedding size/blocks to test the performance of the models and then find $d^*$ or $T^*$ corresponding to the best performance. Then, we select the values near $d^*$ or $T^*$ and show their corresponding results.

### 4.2 DISTINGUISH FOLDABLE INSTANCES

The MILP graphs can be divided into *unfoldable* and *foldable* (Chen et al., 2023b), where foldable MILPs contain many pairs of 1-WL indistinguishable graphs, such as an example in Fig. 2. In this section, we randomly generate 2000 foldable MILPs with 12 variables and 6 constraints, and there are 1000 feasible MILPs while the others are infeasible. Each training set or testing set containing 500 feasible MILPs and 500 infeasible MILPs. Then, we compare the performance of predicting the feasibility of foldable MILPs between our EQGNN and the GNN used in (Chen et al., 2023b) with different numbers of parameters. We set the embedding sizes of GNN as 4, 8, 16, 32, respectively. The number of blocks are set to

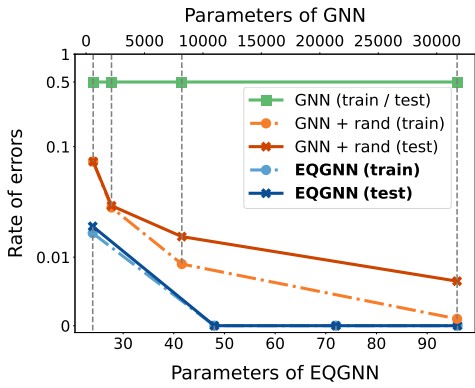

Figure 6: Comparison on foldable MILPs. "GNN + rand": GNN is applied with random features.

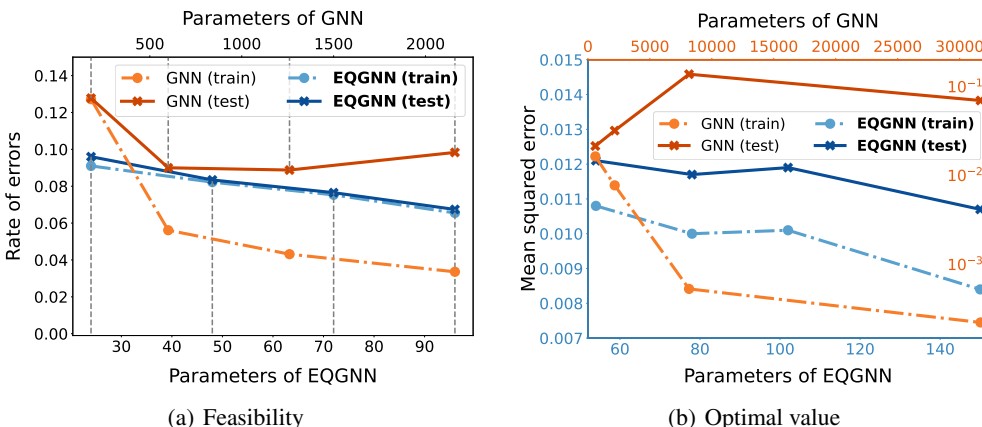

(a) Feasibility            (b) Optimal value

Figure 7: Performance comparison of EQGNN and GNN in predicting feasibility and optimal value of MILPs. GNN exhibits poor generalization performance compared to EQGNN.

$2, 4, 6, 8$. Fig. 6 shows that GNN fails to distinguish the feasibility of foldable MILPs, regardless of the size of GNN. To handle the foldable MILPs, Chen et al. (2023b) proposed to append random features to the MILP-induced graphs, and we also compare the performance of GNN in this case, as shown in orange lines of the figure. Although GNN with random feature can improve the performance, it achieves the best when the embedding size is 32, which will cost $30, 565$ parameters. In contrast, EQGNN can achieve accurate test results with just $48$ parameters, i.e., 4 blocks. The results verify the superiority of EQGNN in both accuracy and the number of model parameters for foldable MILPs.

## 4.3 Experiments on unfoldable MILP

Although GNN cannot distinguish foldable MILPs, it is still capable of distinguishing and representing unfoldable MILPs (Chen et al., 2023b). Therefore, we compare the ability of EQGNN and GNN to predict feasibility, objective and solution on unfoldable instances. We randomly generate $8, 290$ unfoldable MILPs with 4 variables and 4 constraints, where feasible MILPs and infeasible MILPs account for one-half, respectively. The dataset is then equally divided into the train set and test set.

**Feasibility and Optimal value.** As shown in Fig. 7, the performance of predicting the feasibility and objective of unfoldable MILPs is compared between EQGNN and GNN with different parameter counts. For predicting feasibility, the embedding size of the GNN is set as $2, 4, 6, 8$, and the number of blocks of EQGNN is set as $2, 4, 6, 8$. Moreover, since the problem of predicting the optimal value is more complicated, the embedding size of the GNN is set as $4, 8, 16, 32$, and the number of blocks of EQGNN is set as $4, 6, 8, 12$. From the result, we can see that although GNN has better train error as the number of parameters increases, the generalization error increases gradually, such that almost all of the results on the test set are worse than our EQGNN. This means that EQGNN can utilize fewer parameters to achieve better test results and generalization performance.

**Optimal solution.** Considering the large gap between GNN's train and test results, we then compare the ability to approximate optimal solutions by drawing the loss function curve, as illustrated in Fig. 8. For the sake of clarity in the diagram, we only select two hyperparameters of each model for comparison. We trained EQGNN with the number of blocks at $6, 10$, and the number of parameters is $88$ and $148$. The embedding size of GNN is chosen as 8 and 16 with the number of parameters at $2,096$ and $7,904$. We observe that the train performance of GNN increases as the number of parameters increases, but the generalization performance decreases. The train performance of GNN with $d = 8$ is worse than that of EQGNN, and the train performance of GNN with $d = 16$ is better than that of EQGNN. Therefore, we choose the GNNs with these two hyperparameters for comparison. The figure demonstrates that EQGNN has a faster convergence rate and a better generalization performance.

## 4.4 Comparison with other quantum models

Recall Table 3 that most quantum graph neural networks do not consider the feature of the edges. However, the edge features are vital for solving MILP, so we only compare with the QGNN that considered edge features, i.e., quantum graph convolutional neural network (QGCN) (Zheng et al., 2021). In addition, to compare the performance of the problem-agnostic and problem-inspired model, the hardware-efficient ansatz (HEA) (Kandala et al., 2017) is employed. Table 1 reports the rates of

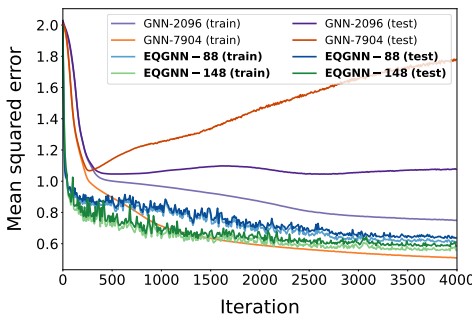

Figure 8: Comparison of train loss and test loss between GNN and EQGNN.

Figure 9: Variance of loss partial derivatives v.s number of qubits n (in log-linear scale).

|  | HEA (Kandala et al., 2017) | QGCN (Zheng et al., 2021) | EQGNN (ours) |
|---|---|---|---|
| **Train** | 0.4613 | 0.3419 | **0.1086** |
| **Test** | 0.4665 | 0.3475 | **0.1127** |

Table 1: Comparison between different quantum models on predicting the feasibility of MILPs, where HEA is problem-agnostic ansatz and QGCN is a quantum graph neural network considering the edge features.

| # Aux. qubits | 0 | 1 | 2 | 3 |
|---|---|---|---|---|
| Train | 0.6580 | 0.6166 | **0.5861** | 0.6299 |
| Test | 0.6853 | 0.6410 | **0.6141** | 0.6554 |

Table 2: Performance change of EQGNN as the number of auxiliary qubits increases on the task of approximating optimal solution of MILPs.

error on predicting the feasibility with different quantum models on an MILP dataset with 3 variables and 3 constraints, which is already close to the limit of our machine simulation QGCN algorithm due to the required number of qubits used by QGCN. In this MILP dataset, QGCN needs 18 qubits while our EQGNN only requires 9 qubits. Moreover, we set the number of parameters of all quantum models as 96 to compare their performance. The result shows that problem-agnostic ansatz cannot effectively learn the separability of samples from graph data. Although QGCN is a problem-inspired ansatz and design equivariant graph convolution layer, their pooling layers break permutation invariance, leading to performance degradation in predicting feasibility of MILP instances. By contrast, our EQGNN can ensure the permutation invariance of the model to achieve better results.

### 4.5 SCALABILITY AND TRAINABLITY

We now study the effect of width of the circuit increased, *i.e.*, using more the number of qubits to solve larger scale problems. A larger circuit width means a larger Hilbert space for the model. However, to maintain equivariance, our model sets the parameter-sharing mechanism, which means that the parameter count within a single block does not increase with the size of the problem. Therefore, to obtain better expressivity for larger problems, a direct way is to increase the number of blocks. In addition, the auxiliary layer in our model is also designed to enhance model expressiveness. By utilizing auxiliary qubits, we can increase both the model's parameters and its capacity while preserving equivariance. Table 2 shows the performance variation of EQGNN in approximating the optimal solution with an increasing number of auxiliary qubits.

When the number of qubits increases, the trainability becomes an issue. It has been shown that generic QNNs suffer from massive local minima (Bittel & Kliesch, 2021) or are prone to barren plateau (McClean et al., 2018), *i.e.*, the loss gradients vanish exponentially with the problem size. Fig. 9 shows the variance of our cost function partial derivatives for a parameter in the middle of the EQGNN. We can see that the variance only decreases polynomially with the system size, which shows the potential of EQGNN to handle larger scale problems.

### 5 CONCLUSION AND OUTLOOK

In this paper, we have presented an Equivariant Quantum Graph Neural Network (EQGNN) approach for solving the MILP problems, to our best knowledge, which has not been devised in literature before. Numerical experiments show that the EQGNN can resolve graphs that GNN cannot distinguish. Moreover, compared with GNN, EQGNN also shows better generalization performance, faster convergence speed, and fewer parameters. The MILP problem can be converted to a weighted bipartite graph with node features, and then predicting feasibility and optimal solution can be regarded as graph classification and regression tasks, which also suggests its broad use as a general quantum neural network for solving more classification and regression problems.

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

## A    RELATED WORK

**Quantum Graph Neural Networks** Various quantum graph neural networks have been presented. (Verdon et al., 2019) proposed a class of graph neural networks by defining operations in terms of Hamiltonians based on the graph structure. However, their models are restricted to Hamiltonians of specific forms, thereby cannot flexibly and efficiently encode classical high-dimensional node or edge features of the graphs to solve some classical tasks. (Zheng et al., 2021) designed a specific quantum graph convolutional neural network (QGCN), which uses amplitude encoding method to encode node and edge features and employs qubits representing edges as control qubits to apply unitaries to the two qubits representing nodes connected by that edge. Nevertheless, the usage of edge qubits will lead to the number of qubits of the model scales quadratically with the number of nodes. Moreover, the pooling layer and measurement operator of QGCN will indeed result in the loss of permutation invariance of the entire model. (Ai et al., 2022) presented an ego-graph based Quantum Graph Neural Network (egoQGNN), which decomposes the input graph into smaller-scale subgraphs and feeds them into the circuit. However, due to the use of entanglement layers within the model, it still does not possess permutation invariance.

**Equivariant Quantum Neural Networks** Recently, a nascent field named geometric quantum machine learning (GQML) (Larocca et al., 2022; Nguyen et al., 2022) has been developed, which leverages the machinery of group and representation theory (Ragone et al., 2022) to build quantum architectures that encode symmetry information about the problem. Schatzki et al. (2022) provide an analytical study of $S_n$-equivariant QNNs and prove that they do not suffer from barren plateaus, quickly reach overparametrization, and can generalize well from small amounts of data. The equivariant QNNs can used to learn various problems with permutation symmetries abound, such as molecular systems, condensed matter systems, and distributed quantum sensors (Peruzzo et al., 2014; Guo et al., 2020), namely, they are also not specifically designed to solve classical graph tasks. (Mernyei et al., 2022) first proposed a theoretical recipe for building permutation equivariant quantum

| Method | Quantum Circuit Embodied | Permutation Invariance | Attribute | Layer | Readout | Application |
|---|---|---|---|---|---|---|
| QGNN (Verdon et al., 2019) | ✗ | ✓ | — | Q | Tomography | Learning Hamiltonian Dynamics & Graph Isomorphism Classification |
| QGCN (Zheng et al., 2021) | ✓ | ✗ | Node & Edge | Q | Estimation | Image Classification |
| egoQGNN (Ai et al., 2022) | ✓ | ✗ | Node | Q & C | Tomography | Graph Classification |
| EQGC (Mernyei et al., 2022) | ✗ | ✓ | Node | Q & C | Estimation | Synthetic Cycle Graph Classification |
| **Ours** | ✓ | ✓ | Node & Edge | Q | Estimation | Graph Classification & Regression |

Table 3: Comparison of different quantum graph neural networks including our method on whether the models provide explicit circuits, exhibit permutation invariance, consider multi-dimensional node or edge features, utilize quantum (Q) layers or classical (C) layers, and the readout manner (tomography or estimation), where the tomography may require an exponentially large number of measurements. Our proposed model shows sound properties many aspects and can be applied to graph classification and regression tasks on real graph problems as also verified in our experiments.

graph circuits (EQGC) and aggregated the output of the quantum circuit by classical functions to ensure permutation invariance of the model. Nevertheless, the EQGC does not provide the specific circuit implementation and does not consider the case of weighted graphs in their model. In addition, another QNN with permutation equivariance (Ye et al., 2023) is proposed, which is specially designed for solving quadratic assignment problems, but their model only encodes the graph information and then employs the shared problem-agnostic ansatz to learn the representation of each node. Thus, their model does not contain the learnable graph message interaction layer.

**Quantum Algorithms for MILP** Mixed-Integer Linear Programming (MILP) is a mathematical optimization approach that aims to find the best solution to a linear objective function while imposing constraints on some or all of the variables to be integers. MILP is widely used in various practical applications such as process scheduling (Floudas & Lin, 2005), transportation (Richards & How, 2002), and network design (Fortz & Poss, 2009). Recently, researchers are endeavoring to employ quantum computing to assist in solving the MILP. (Zhao et al., 2022) proposed a hybrid quantum-classical Benders' decomposition algorithm, which decomposes an MILP problem into a Quadratic unconstrained binary optimization (QUBO) problem solved by quantum computer and a subproblem easily tackled by classical computers. (Ossorio-Castillo & Pena-Brage, 2022) described a algorithm based on Dantzig–Wolfe decomposition. Different from (Zhao et al., 2022), the algorithm then solve several either continuous or binary subproblems instead of a mixed one. (Wang et al., 2022b) pointed out that quantum-inspired Ising machines can be used to solve MILPs by reducing them into Ising models. However, the above algorithms are based on unconstrained Ising models, while MILPs subject to complex constraints. Their common solution is to introduce a penalty to the algorithm. A proper penalty is of great importance because an extremely large penalty may cause quantum annealer malfunctioning since it will explode the coefficients while a soft penalty may make quantum annealer ignore the corresponding constraints (Zhao et al., 2022). However, there is no instruction on how to tune the penalty and it may even be different to various MILP problems. In contrast, our approach leverages QML to represent MILP problems, thereby pioneering a novel direction for harnessing quantum computing in aiding MILP solutions, there is promising for witnessing the emergence of new paradigms that combine quantum and classical methods for MILP solving.

**Remark.** Equivariant Graph Neural Networks have garnered significant attention, especially in their applications for handling molecular systems within 3D spaces (Liu et al., 2022; Hoogeboom et al., 2022). Within the domain of computational biology, the concept of equivariance, as related to the inherent symmetries of molecular systems, postulates the following: when spatial transformations, such as rotations or translations, are applied to the input atomic coordinates, the output coordinates produced by the network should also manifest these transformations in a congruent manner (Satorras et al., 2021; Gasteiger et al., 2021). However, in the context of this paper, our definition of equivariance is specifically oriented towards the notion that the output of the graph neural network maintains an equivariant relationship with respect to the sequential arrangement of nodes in the input graph.

## B EQGNN CAN DISTINGUISH GNN INDISTINGUISHABLE MILP GRAPHS

The separation power of GNN is measured by whether it can distinguish two non-isomorphic graphs (Chen et al., 2023b) and it has been shown that GNN has the same separation power with the WL

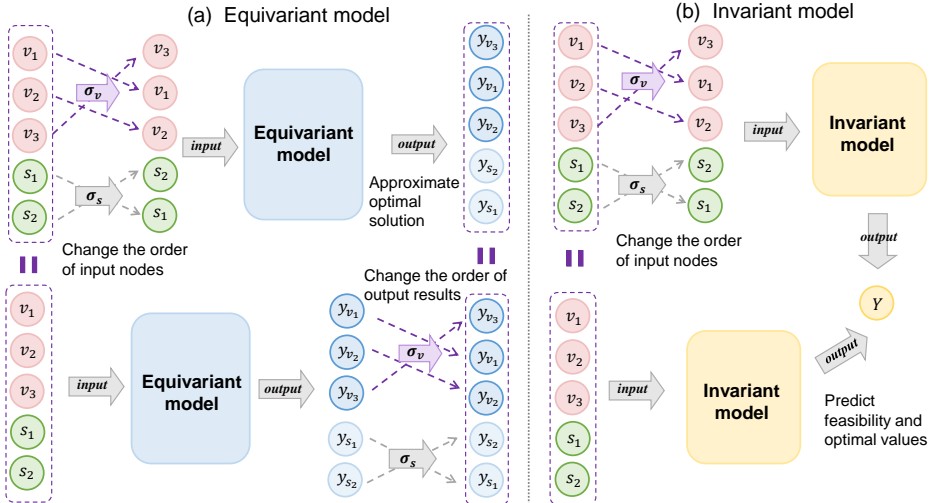

Figure 10: Diagram of the properties of equivariant (a) and invariant (b) models. For equivariant models, when the input node has a permutation $\sigma$, the output is equivalent to the original one with the same permutation $\sigma$. For invariant models, the permutation of input nodes will not affect output $Y$.

---

**Algorithm 1:** WL test for MILP-Graphs

---

**Input:** A graph instance $(G, H) \in \mathcal{G}_{q,p} \times \mathcal{H}_q^V \times \mathcal{H}_p^S$, and iteration limit $L > 0$.

1  Initialize with $c_v^0 = hash(\boldsymbol{f}^V)$ for all $v \in V$, $c_s^0 = hash(\boldsymbol{f}^S)$ for all $s \in S$.

2  **foreach** $l = 1, 2, ... L$ **do**

3       $c_{v_i}^l = hash(c_{v_i}^{l-1}, \sum_{j=0}^{p-1} A_{i,j} hash(c_{s_j}^{l-1}))$, for all $v \in V$

4       $c_{s_j}^l = hash(c_{s_j}^{l-1}, \sum_{i=0}^{q-1} A_{i,j} hash(c_{v_i}^{l-1}))$, for all $s \in S$

5  **end**

**Result:** The multisets containing all colors $\{\{c_v^l : v \in V, c_s^l : s \in S\}\}$

---

test (Xu et al., 2018). Therefore, we first demonstrate why 1-WL test and GNNs fail to distinguish between some non-isomorphic graphs (i.e., components of the foldable dataset), and then demonstrate how our EQGNN distinguishes between them, and thus show that EGNN can surpass the separation power of GNNs. We present a variant of WL test specially modified for MILP that follows the same lines as in Chen et al. (2023b).

In the algorithm, $hash()$ is a function that maps its input feature to a color in $\mathcal{C}$. The algorithm flow can be seen as follows. First, all nodes in $V$ sup $S$ are assigned to an initial color $c_v^0$ and $c_s^0$ according to their node features. Then, for each $v_i \in V$, $hash$ function maps the previous color $c_{v_i}^{l-1}$ and aggregates the color of the neighbors of $\{c_{s_j}^{l-1}\}_{s_j \in \mathcal{N}(v_i)}$. Similarly, $hash$ function maps the previous color $c_{s_j}^{l-1}$ and aggregates the color of the neighbors of $\{c_{v_i}^{l-1}\}_{v_i \in \mathcal{N}(s_j)}$. This process is repeated until $L$ reaches the maximum iteration number. Finally, a histogram $h_G$ of the node colors can be obtained according to $\{\{c_v^l : v \in V, c_s^l : s \in S\}\}$, which can be used as a canonical graph representation. The notation $\{\{\cdot\}\}$ denotes a multiset, which is a generalisation of the concept of set in which elements may appear multiple times: an unordered sequence of elements. That is, the 1-WL test transforms a graph into a canonical representation, and then if the canonical representation of two graphs is equal, 1-WL test will consider them isomorphic.

Next, we show a case where the 1-WL test fails. Taking Fig. 2 as an example, we feed (a) and (b) from Fig. 2 into this algorithm separately and compare the differences at each step. Initially, each variable vertex has the same color $c_v^0$, and each constraint vertex has the same color $c_s^0$, because their features are the same. This step is the same for Fig. 2(a) and Fig. 2(b). Then, each vertex iteratively updates its color, based on its own color and information from its neighbors. Since each $v_i$ is connected to two $s_{j_1}, s_{j_2}$ in both (a) and (n), they will also obtain the consistent $c_v^l$ and $c_s^l$ at this step. That is, (a) and (b) will get the same representation by the algorithm, and they cannot be distinguished by 1-WL test GNNs. Moreover, Xu et al. (2018) show that GNNs are at most as powerful as the

1-WL test in distinguishing graph structures. Therefore, a pair of 1-WL indistinguishable graphs are indistinguishable by GNNs.

For EQGNN, the subfigure (a) and (b) of Fig. 2 can be clearly distinguished. Since the two graphs differ only in the connectivity of edges, we only consider the modules associated with edges in EQGNN, i.e., $U_{g_1}$ and $U_{g2}$. In subfigure (a) node $v_3$ is connected to $s_3$ and $s_4$, while in subfigure (b) node $v_3$ is connected to $s_3$ and $s_1$. Therefore, the difference between $U_{g_1}(G_a, \beta)$ and $U_{g_1}(G_b, \beta)$ includes $\exp(-\mathbf{i}((A_{3,4} + \beta)Z_{2*3}Z_{2*6+4}))$ and $\exp(-\mathbf{i}((A_{3,1} + \beta)Z_{2*3}Z_{2*6+1}))$. Since the edge weights and the parameter $\beta$ are equal, we can just compare $\exp(-\mathbf{i}(Z_6 Z_{13}))$ and $\exp(-\mathbf{i}(Z_6 Z_{16}))$. These two values are obviously not equal, so $U_{g_1}(G_a, \beta) \neq U_{g_1}(G_b, \beta)$ and the overall model $U(G_a, H) \neq U(G_b, H)$. Thereby, EQGNN can distinguish such graphs that the 1-WL test and GNNs cannot.

## C  THE SEPARATION POWER OF EQGNN

**Theorem 4.** *EQGNN can encode the two different graphs to different representation.*

Suppose there are two graphs $\mathcal{G}_1 = (G_1, H_1) = (V_1 \cup S_1, A_1, H_1)$ and $\mathcal{G}_2 = (G_2, H_2) = (V_2 \cup S_2, A_2, H_2)$, and the differences between the two graphs may appear in three places, i.e., the features of nodes, the weight and connectivity of edges. As we mentioned in Definition 4, there are two types of layers in $U_{\boldsymbol{\theta}}(G, H)$, one is independent of input graph $(G, H)$, and the other is related to input graph $(G, H)$. In the proposed EQGNN, only $U_{x_1}(c, u, b)$ and $U_{x_3}(l, \epsilon, \circ)$ are related to the features of nodes. And $U_{g_1}(G, \beta_t)$ and $U_{g_2}(G, \beta_t)$ are related to the weight and connectivity of edges.

For $A_1 = A_2, H_1 \neq H_2$, the only difference between two quantum circuits is $U_{x_1}(c, u, b)$ and $U_{x_2}(l, \epsilon, \circ)$. Taking $U_{x_1}(c, u, b)$ for example, $U_{x_1}(c, u, b) = \exp(-\mathbf{i}(\sum_{i=0}^{q-1}(c_i X_{2i} + u_i X_{2i+1}) + \sum_{j=0}^{p-1} b_j X_{2q+j}))$. If any feature changes, $U_{x_1}$ will change, causing $U_{\boldsymbol{\theta}}(G_1, H_1)$ and $U_{\boldsymbol{\theta}}(G_2, H_2)$ to be different.

For $A_1 \neq A_2, H_1 = H_2$, there are two cases where the connections of the edges are different or the weights of the edges are different. We have already demonstrated the former in Appendix B, and now we focus on the latter. $U_{g_1}(G, \beta) = \exp(-\mathbf{i}(\sum_{(i,j)\in\mathcal{E}}(A_{i,j} + \beta)Z_{2i}Z_{2q+j}))$. The change of $A_{i,j}$ will lead the change of $U_{g_1}(G, \beta)$, causing $U_{\boldsymbol{\theta}}(G_1, H_1)$ and $U_{\boldsymbol{\theta}}(G_2, H_2)$ to be different.

The discussion on $A_1 \neq A_2, H_1 \neq H_2$ as follows. As we can see, in the feature encoding layer, we use the $\exp(-\mathbf{i}f(X))$ gate to encode features, and the graph information interaction uses $\exp(-\mathbf{i}b(Z_i Z_j))$, where $f \in H$ denotes features and $b \in A$ denotes edge weights. We specially use different Pauli gates ($X$ and $Z$) as the bases of the two layers, so as to ensure that as long as one of the layers changes, the whole unitary changes.

## D  DATASET GENERATION

Foldable dataset is constructed by many pairs of 1-WL indistinguishable graphs, and Fig. 2 in our paper is a foldable example, which is a pair of non-isomorphic graphs that cannot be distinguished by WL-test or by GNNs, while unfoldable dataset refers to other MILP instances that do not have 1-WL indistinguishable graphs.

In section 4.2, we randomly generate 2000 foldable MILPs with 12 variables and 6 constraints, and there are 1000 feasible MILPs with attachable optimal solution while the others are infeasible. We construct the $(2k-1)$-th and $2k$-th problems via following approach ($1 \leq k \leq 500$).

- Sample $J = \{j_1, j_2, ..., j_6\}$ as a random subset of $\{1, 2, ..., 12\}$ with 6 elements. 1. For $j \in J$, $x_j \in \{0, 1\}$, i.e., $x_j$ is a binary integer variable. 2. For $j \notin J$, $x_j$ is a continuous variable with bounds $l_j \sim \mathcal{U}(0, \pi), u_j \sim \mathcal{U}(0, \pi)$. If $l_j > u_j$, then switch $l_j$ and $u_j$.

- $c_1 = ... = c_{12} = 0.01$.

- The constraints for the $(2k-1)$-th problem (feasible) is $x_{j_1} + x_{j_2} = 1, x_{j_2} + x_{j_3} = 1,$ $x_{j_3} + x_{j_4} = 1, x_{j_4} + x_{j_5} = 1, x_{j_5} + x_{j_6} = 1, x_{j_6} + x_{j_1} = 1$. For example, $x = (0, 1, 0, 1, 0, 1)$ is a feasible solution.

- The constraints for the $2k$-th problem (infeasible) is $x_{j_1} + x_{j_2} = 1$, $x_{j_2} + x_{j_3} = 1$, $x_{j_3} + x_{j_1} = 1$, $x_{j_4} + x_{j_5} = 1$, $x_{j_5} + x_{j_6} = 1$, $x_{j_6} + x_{j_4} = 1$. An explanation for why it is infeasible: we add the first three formulas together to get $2(x_{j_1} + x_{j_2} + x_{j_3}) = 3$, but $x_j \in \{0, 1\}$ for $j \in J$, so there will be no solution that satisfies this equation and the MILP problem must be infeasible.

For unfoldable MILPs, we first set the number of variables and constraints to $m$ and $n$.

- For each variable, $c_j \sim \mathcal{N}(0, 0.01)$, $l_j, u_j \sim \mathcal{N}(0, \pi)$. If $l_j > u_j$, then switch $l_j$ and $u_j$. The probability that $x_j$ is an integer variable is 0.5.
- For each constraint, $\circ_i \sim \mathcal{U}(\leq, =, \geq)$ and $b_i \sim \mathcal{U}(-1, 1)$.
- After randomly generating all the MILP samples, we use the 1-WL test algorithm to calculate their graph representation for each instance, ensuring that there are no duplicate graph representations in the dataset, so that we can determine that this dataset does not contain 1-WL indistinguishable pairs of MILP instances.

# E  THE EFFECTIVE DIMENSION OF EQGNN

Abbas et al. (2021) introduce the effective dimension as a useful indicator of how well a particular model will be able to perform on the dataset. And we use the presented tool to quantify the expressiveness of our model. In particular, this algorithm follows 4 main steps:

1) Monte Carlo Simulation: the forward and backward passes (gradients) of the neural network are computed for each pair of input and weight samples.

2) Fisher Matrix Computation: these outputs and gradients are used to compute the Fisher Information Matrix.

$$\tilde{F}_k(\theta) = \frac{1}{k} \sum_{j=1}^{k} \frac{\partial}{\partial \theta} \log p(x_j, y_j; \theta) \frac{\partial}{\partial \theta} \log p(x_j, y_j; \theta)^{\mathsf{T}},$$

where $(x_j, y_j)_j^k = 1$ are i.i.d. drawn from the distribution $p(x, y; \theta)$.

3) Fisher Matrix Normalization: averaging over all input samples and dividing by the matrix trace.

4) Effective Dimension Calculation: according to the formula

$$d_{\gamma,n}(\mathcal{M}_\Theta) := 2 \frac{\log \left( \frac{1}{V_\Theta} \int_\Theta \sqrt{\det \left( \mathrm{id}_d + \frac{\gamma n}{2\pi \log n} \hat{F}(\theta) \right)} d\theta \right)}{\log \left( \frac{\gamma n}{2\pi \log n} \right)},$$

where $\mathcal{M}_\Theta$ is a statistical model with $\gamma \in (0, 1]$, $\Theta \in \mathbb{R}^d$ and data samples $n \in \mathbb{N}$. $V_\Theta := \int_\Theta d\theta \in \mathbb{R}_+$ is the volume of the parameter space. $\hat{F}_{ij} \in \mathbb{R}^{d \times d}$ is the normalised Fisher information matrix defined as

$$\hat{F}_{ij}(\theta) := d \frac{V_\Theta}{\int_\Theta \mathrm{tr}(F(\theta)) d\theta} F_{ij}(\theta).$$

We test the proposed model with two blocks on the MILP dataset and use the random weight to evaluate the normalized effective dimension. As shown in Fig. 11, with the increase in the number of data, the normalized effective dimension of the proposed model can converge to near 0.75, which shows that our model can achieve good expressiveness even with two blocks.

# F  PROOF OF THEOREM 2

*Proof.* $U_{x_1}(c, u, b) = \bigotimes_{i=0}^{q-1} [\mathrm{R_x}(c_i) \otimes \mathrm{R_x}(u_i)] \otimes [\bigotimes_{j=0}^{p-1} \mathrm{R_x}(b_j)]$, where $c$ and $u$ are the features of variables, and $b$ is the feature of constraints. After the order of the input nodes is changed, $U_{x_1}(\sigma_v(c, u), \sigma_s(b)) = \bigotimes_{i=0}^{q-1} [\mathrm{R_x}(c_{\sigma_v(i)}) \otimes \mathrm{R_x}(u_{\sigma_v(i)})] \otimes [\bigotimes_{j=0}^{p-1} \mathrm{R_x}(b_{\sigma_s(j)})]$. The representation $R(\sigma_v)$ can be implemented by a sequence of SWAP gates, swapping the $2i$-th qubit for the $2\sigma_v(i)$-th qubit and swapping the $(2i + 1)$-th qubit for the $(2\sigma_v(i) + 1)$-th qubit. The representation $R(\sigma_s)$ can

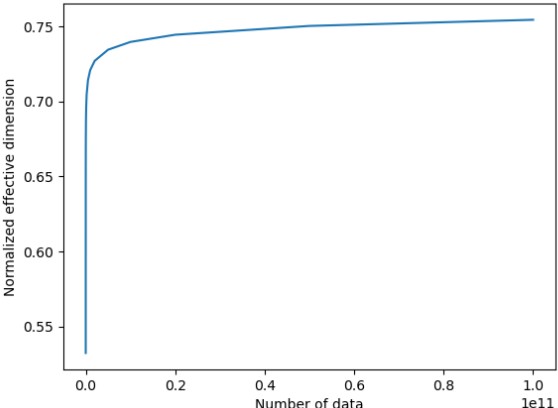

Figure 11: The effective dimension of EQGNN. With the increase in the number of data, the normalized effective dimension of the proposed model can converge to near 0.75.

also be implemented by a sequence of SWAP gates, swapping the $j$-th qubit for the $\sigma_s(j)$-th qubit. Such that $(R(\sigma_v) \otimes R(\sigma_s))U_{x_1}(c, u, b) = \bigotimes_{i=0}^{q-1} \left[ \mathrm{R_x}(c_{\sigma_v(i)}) \otimes \mathrm{R_x}(u_{\sigma_v(i)}) \right] \otimes [\bigotimes_{j=0}^{p-1} \mathrm{R_x}(b_{\sigma_s(j)})] = U_{x_1}(\sigma_v(c, u), \sigma_s(b))$. Thereby, $U_{x_1}(c, u, b)$ is equivariant w.r.t. the permutations $\sigma_v$ and $\sigma_s$. $U_{x_3}(l, \epsilon, \circ) = \bigotimes_{i=0}^{q-1} \left[ \mathrm{R_z}(l_i) \otimes \mathrm{R_z}(\epsilon_i) \right] \otimes [\bigotimes_{j=0}^{p-1} \mathrm{R_z}(\circ_j)]$, which has a similar structure and can be similarly proven to be equivariant. □

## G   FORMULAS

Quantum circuits comprise quantum gates. Some commonly used single-qubit gates include the Pauli-X gate, the Pauli-Y gate, and the Pauli-Z gate, which correspond to rotations of $\pi$ around the x, y, and z axes on the Bloch sphere, respectively. Parametric quantum circuits (PQCs) mean the used quantum gates are usually parameterized, namely, the gates contain learnable parameters, e.g., $\mathrm{R_x}(\theta)$, $\mathrm{R_y}(\theta)$, $\mathrm{R_z}(\theta)$, $\mathrm{R_{zz}}(\theta)$. We can use classical optimizers to minimize a cost function by adjusting the parameters of quantum gates. The cost is evaluated by applying the PQC to a set of input states and measuring the output probabilities and is typically chosen to be related to the objective function of the optimization task. The common parametric quantum gates include

$$\mathrm{R_x}(\theta) = \begin{bmatrix} \cos\left(\frac{\theta}{2}\right) & -i\sin\left(\frac{\theta}{2}\right) \\ -i\sin\left(\frac{\theta}{2}\right) & \cos\left(\frac{\theta}{2}\right) \end{bmatrix} \qquad \mathrm{R_y}(\theta) = \begin{bmatrix} \cos\left(\frac{\theta}{2}\right) & -\sin\left(\frac{\theta}{2}\right) \\ \sin\left(\frac{\theta}{2}\right) & \cos\left(\frac{\theta}{2}\right) \end{bmatrix} \tag{9}$$

$$\mathrm{R_z}(\theta) = \begin{bmatrix} e^{-i\frac{\theta}{2}} & 0 \\ 0 & e^{i\frac{\theta}{2}} \end{bmatrix} \qquad \mathrm{R_{zz}}(\theta) = \begin{bmatrix} e^{-i\frac{\theta}{2}} & 0 & 0 & 0 \\ 0 & e^{i\frac{\theta}{2}} & 0 & 0 \\ 0 & 0 & e^{i\frac{\theta}{2}} & 0 \\ 0 & 0 & 0 & e^{-i\frac{\theta}{2}} \end{bmatrix}. \tag{10}$$

| Embedding size ($d$) | 1 | 2 | 3 | 4 | 5 | 6 | 7 | 8 | 9 | 10 | 11 | 12 |
|---|---|---|---|---|---|---|---|---|---|---|---|---|
| # Param. | 36 | 104 | 204 | 336 | 500 | 696 | 924 | 1184 | 1476 | 1800 | 2156 | 2544 |
| Train_err | 0.5 | 0.1216 | 0.0882 | 0.0787 | 0.0627 | 0.0635 | 0.0534 | 0.0436 | 0.0235 | 0.008 | 0.007 | 0.003 |
| Test_err | 0.5 | 0.1285 | 0.1062 | 0.1057 | 0.0924 | 0.0991 | 0.0983 | 0.1133 | 0.0950 | 0.0821 | 0.0897 | 0.0888 |
| Generalization error | 0 | 0.0069 | 0.0180 | 0.0270 | 0.0297 | 0.0356 | 0.0449 | 0.0697 | 0.0715 | 0.0741 | 0.0827 | 0.0858 |

Table 4: Performance of GNNs with 1 layer at different embedding sizes on the predicting feasibility on the unfoldable dataset.

| # Blocks ($T$) | 1 | 2 | 3 | 4 | 5 | 6 | 7 | 8 | 9 | 10 |
|---|---|---|---|---|---|---|---|---|---|---|
| # Param. | 12 | 24 | 36 | 48 | 60 | 72 | 84 | 96 | 108 | 120 |
| Train_err | 0.0988 | 0.091 | 0.0868 | 0.0822 | 0.0757 | 0.0733 | 0.0707 | 0.0655 | 0.0632 | 0.0618 |
| Test_err | 0.1001 | 0.096 | 0.0913 | 0.0834 | 0.0768 | 0.0745 | 0.0721 | 0.0674 | 0.0655 | 0.0634 |
| Generalization error | 0.0013 | 0.005 | 0.045 | 0.0012 | 0.0011 | 0.0012 | 0.0014 | 0.0019 | 0.0021 | 0.0016 |

Table 5: Performance of EQGNN with different numbers of blocks on the predicting feasibility on the unfoldable dataset.

The feature encoding layer can be seen as being made of four layers, *i.e.*, $U_x^t(G, H, \alpha_t) = U_{x_1}(c, u, b) \cdot U_{x_2}(\alpha_t) \cdot U_{x_3}(l, \epsilon, \circ) \cdot U_{x_4}(\alpha_1)$, where:

$$
\begin{aligned}
U_{x_1}(c, u, b) =& \exp\left(-\mathbf{i}\left(\sum_{i=0}^{q-1}(c_i X_{2i} + u_i X_{2i+1}) + \sum_{j=0}^{p-1} b_j X_{2q+j}\right)\right) \\
U_{x_2}(\alpha_t) =& \exp\left(-\mathbf{i}\left((\sum_{i=0}^{q-1}(\alpha_{t,1} X_{2i} + \alpha_{t,2} X_{2i+1}) + \sum_{j=0}^{p-1} \alpha_{t,5} X_{2q+j}\right)\right) \\
U_{x_3}(l, \epsilon, \circ) =& \exp\left(-\mathbf{i}\left(\sum_{i=0}^{q-1}(l_i X_{2i} + \epsilon_i X_{2i+1}) + \sum_{j=0}^{p-1} \circ_j X_{2q+j}\right)\right) \\
U_{x_4}(\alpha_t) =& \exp\left(-\mathbf{i}\left(\sum_{i=0}^{q-1}(\alpha_{t,3} X_{2i} + \alpha_{t,4} X_{2i+1}) + \sum_{j=0}^{p-1} \alpha_{t,6} X_{2q+j}\right)\right).
\end{aligned}
\tag{11}
$$

## H  NUMERICAL EXPERIMENTS

In this section, we provide more numerical experiments to show the performance change of GNN and EQGNN as the number of parameters increases. As shown in Table 4, the train errors of GNN decrease as the parameter counts increase, but the generalization errors increase as the number of parameters increases. Moreover, the test errors of GNN stay stable at around $0.09$. In contrast, the test and train errors of EQGNN can decrease as the parameter counts increase, and the generalization errors are always small, such that EQGNN can achieve better results than GNN on the unfoldable dataset. In addition, we can compare the GNN with $d = 1$ with EQGNN with $T = 3$, because they have the same number of parameters. In this case, EQGNN can achieve better results than GNN in both training and testing. Similarly, comparing GNN with $d = 2$ with EQGNN with $T = 9$, we can obtain similar results. Furthermore, as we can see, only when GNN has more than 696 parameters does its train performance exceed EQGNN with 120 parameters. However, GNN with 696 parameters has poor generalization performance, such that the test performance of GNN is much lower than that of EQGNN.

