# OpenReview forum: "Equivariant Quantum Graph Neural Network for Mixed-Integer Linear Programming"
_ICLR.cc/2024/Conference — Submitted to ICLR 2024_

### Official Review · Reviewer_43Zw · 2023-10-19

**Soundness:** 3 good
**Presentation:** 3 good
**Contribution:** 2 fair
**Rating:** 6
**Confidence:** 3

**Summary:**

The authors introduce a class of quantum-based graph neural networks equivariant under permutations of the vertices of the graph. The authors argue that this class of neural networks are effective in solving mixed-integer linear programming (MILP) problems, and back up this intuition with numerics. They also check in their numerics the trainability of their introduced class of quantum neural networks.

**Strengths:**

The authors study the trainability of their model; due to its large symmetry group (permutation invariance), the authors believe the model is trainable (when there are few auxiliary qubits) and, indeed, demonstrate it empirically. The authors never state it, but I believe this is essentially due to the results of arXiv:2210.09974, which give trainability guarantees for permutation invariant quantum machine learning models.

**Weaknesses:**

I am more skeptical of the numerical results demonstrating a separation in expressive power between the EQGNN and the GNN the authors cite from (Chen et al., 2023b) (though admittedly I am not an expert on GNNs). The divergence of testing performance in Fig. 10 seems to me that the classical GNN is overfitting, potentially due to the order of magnitude difference in the numbers of parameters between the quantum and classical models. I highly recommend the authors perform supplemental numerics where these parameter counts are brought in line to control for this behavior. It is also unclear to me whether there is actually any theoretical quantum advantage when the quantum model has no auxiliary qubits as arXiv:2211.16998 gives efficient classical simulation algorithms for permutation-invariant quantum systems and machine learning models. This might limit the utility of the introduced quantum model to the case where there are many auxiliary qubits, which then runs into problematic training behavior as the authors point out.

A much more minor point, but there are also many typos: "and yielding" at the bottom of page 3, "expressivity power" in the abstract, "guarantee to distinguish" in the abstract, and "TorchQauntum" at the bottom of page 7 are some that I found.

**Questions:**

What are the connections between this work and previous work on permutation-invariant quantum machine learning models (cited above)? What explains the terrible generalization performance of the classical GNNs? The authors should perform additional numerics fixing the parameter counts of the classical and quantum models to control for this behavior.

---

> ### Author Response · Authors · 2023-11-19
> **Response to Reviewer 43Zw (Part 1)**
>
> >***W1: I am more skeptical of the numerical results demonstrating a separation in expressive power between the EQGNN and the GNN the authors cite from (Chen et al., 2023b) (though admittedly I am not an expert on GNNs). The divergence of testing performance in Fig. 10 seems to me that the classical GNN is overfitting, potentially due to the order of magnitude difference in the numbers of parameters between the quantum and classical models. I highly recommend the authors perform supplemental numerics where these parameter counts are brought in line to control for this behavior.***
>
> **A1:** Thanks for your suggestion. Next, we add more numerical experiments to explain this problem. But first, we need to clarify that the number of parameters of the GNN is limited by its structure. That is, we cannot guarantee that the number of parameters in the GNN is exactly linear. More specifically, the GNNs used in (Chen et al., 2023b) are constructed by the following structure.
>
> (1) The initial embedded features are $s_i^0 = f_{in}^V(h_i^V)$ and $t_j^0 = f_{in}^W(h_j^W)$, where $V$ represents the the set of variable nodes, and $W$ represents the the set of constraint nodes. $h_i^V$ indicates the input features of variable nodes, and $h_j^W$ indicates the input features of constraint nodes.  $f_{in}^V: \mathbb R^{d_{in}} \rightarrow \mathbb R^{d}$, where $d_{input}$ is the dimension of input features and $d$ is defined as the **embedding size**. $f_{in}^V$ and $f_{in}^W$ are the learnable functions, which are parameterized with multilayer perceptrons (MLPs), and have two hidden layers with a ReLU activation function. The number of parameters of $f_{in}^V$ is calcuated by $2*(d_{in}*d+d)$.
>
> (2) Message-passing layers for $l = 1, 2, . . . , L$. Given learnable mappings
> $\\{f_l^V , f_l^W , f_l^V , f_l^W, f_{l_{out}}^V , f_{l_{out}}^W  \\}_{l=0}^L$,
> one can update vertex features by the following formulas:
>
> $$s_i^l = f_{l_{out}}^V(f_l^V(s_i^{l-1}), \sum_{j=1}^n E_{i,j} f_l^W(t_j^{l-1})), \quad t_j^l = f_{l_{out}}^W(f_l^W(t_j^{l-1}),\sum_{i=1}^m E_{i,j} f_l^V(s_i^{l-1})),$$
> where all learnable mappings $f$ are also parameterized with multilayer perceptrons (MLPs) and have two hidden layers with a ReLU activation function.
>
> (3) For predicting feasibility, the output is $f_{out} (\sum_{i=0}^n{s_i^L}, \sum_{j=0}^n{t_i^L})$,
> $f_{out}: \mathbb R^{d} \rightarrow \mathbb R$, and the number of parameters is $2*d^2 + 2d$.
>
> Therefore, the number of parameters of the GNN is  $(14L+2)d^2 + (12L+8)d$ in the case where $h_i^V \in \mathbb R^4$ and $h_j^W \in \mathbb R^2$. In the paper (Chen et al., 2023b), they set $L = 2$ for the GNN. Note that due to the over-smoothing of GNNs, most GNNs only stack two to four layers, i.e., $L = 2 \sim 4.$ Therefore, the best way to control performance is to vary the embedding size $d$. When $L=2$, the number of parameters of the GNN is $30d^2+30d$. Since $d \in \mathbb Z^+$, the number of parameters of GNN can only take $60, 180, 360, 600, 900,...$.
>
>
> Our model uses the number of blocks $T$ to control the number of parameters, and each block has 12 parameters, so the number of our parameters is $12*T$. We converted Figure 7(a) of our paper into a table form and attached the embedding size of the GNN and the number of blocks of our EQGNN, as shown below.
>
> The results of the GNN:
> |                      |                     |                     |                      |                     |
> |:--------------------:|:-------------------:|:-------------------:|:--------------------:|:-------------------:|
> |     **Embedding Size ($d$)**      |          **2**          |          **4**          |          **6**           |          **8**          |
> |     **# Parameters**      |          180          |          600          |          1260           |          2160          |
> | **Train error rate** | 0.1272 | 0.0561 | 0.0431 | 0.0336 |
> | **Test error rate**  | 0.1278 | 0.0900 | 0.0888  | 0.0983 |
>
> The results of our EQGNN:
> |                      |        |        |        |        |
> |:--------------------:|:------:|:------:|:------:|:------:|
> | **# Blocks ($T$)**         | **2**      | **4**      | **6**      | **8**      |
> | **# Parameters**         | 24      | 48      | 72      | 96      |
> | **Train error rate** | 0.0910 | 0.0822 | 0.0753 | 0.0655 |
> | **Test error rate**  | 0.0960 | 0.0834 | 0.0765 | 0.0674 |
>
> As we can see, the purpose of this experiment is to control the embedding size of the GNN and the number of blocks of EQGNN to be consistent. In all our experiments, we first gradually increase the embedding size/blocks to test the performance of the models and then find the embedding size/blocks $d^*$ or $T^*$ corresponding to the best performance. Then, we select the values near $d^*$ or $T^*$ and show their corresponding results in the experiments of the paper.

---

> ### Author Response · Authors · 2023-11-19
> **Response to Reviewer 43Zw (Part 2)**
>
> However, in order to further study the performance of GNN and further reduce the number of GNN parameters, we set the layer of GNN $L=1$, and the number of parameters is calculated by $16d^2+20d$. Note that the number of parameters when $L=1$ and $d=1$ is already the minimum number of parameters for this model. Then, we vary $d$ from 1 to 12 to get the following result:
>
> |                          |     |        |        |        |        |        |        |        |        |        |        |        |
> |:------------------------:|:---:|:------:|:------:|:------:|:------:|:------:|:------:|:------:|:------:|:------:|:------:|:------:|
> | **Embedding size ($d$)** |  1  |   2    |   3    |   4    |   5    |   6    |   7    |   8    |   9    |   10   |   11   |   12   |
> |      **Parameters**      | 36  |  104   |  204   |  336   |  500   |  696   |  924   |  1184  |  1476  |  1800  |  2156  |  2544  |
> |   **Train error rate**   | 0.5 | 0.1216 | 0.0882 | 0.0787 | 0.0627 | 0.0635 | 0.0534 | 0.0436 | 0.0235 | 0.008  | 0.007  | 0.003  |
> |   **Test error rate**    | 0.5 | 0.1285 | 0.1062 | 0.1057 | 0.0924 | 0.0991 | 0.0983 | 0.1133 | 0.0950 | 0.0821 | 0.0897 | 0.0888 |
> |**Generalization error**  |  0  | 0.0069 | 0.0180 | 0.0270 | 0.0297 | 0.0356 | 0.0449 | 0.0697 | 0.0715 | 0.0741 | 0.0827 | 0.0858 |
>
> As we can see, with the increase in embedding size, the training error of the model decreases continuously, but the generalization errors increase. Moreover, the test errors of GNN stay stable at around $0.09$, which indicates that GNN does not generalize well on the MILP dataset. As a comparison, we vary the number of blocks $T$ to test the performance of EQGNN, i.e.,
>
> | # Blocks ($T$)       | 1    | 2    | 3   | 4   | 5   | 6   | 7   | 8   | 9   | 10  |
> | -------------------- | ---- | ---- | --- | --- | --- | --- | --- | --- | --- | --- |
> | **# Parameters**     |  12    |  24    |  36   |  48   |  60   |  72   |  84   |  96   |  108   | 120 |
> | **Train error rate** |  0.0988 |  0.091 | 0.0868 | 0.0822  | 0.0757  |  0.0733   | 0.0707 | 0.0655 | 0.0632| 0.0618 |
> | **Test error rate**  |  0.1001 |  0.096 |  0.0913  |  0.0834  |  0.0768  |  0.0745  |  0.0721 |  0.0674 |  0.0655 |  0.0634|
> | **Generalization error**   |0.0013  | 0.005 |0.045 | 0.0012 | 0.0011  |0.0012 | 0.0014 |0.0019   | 0.0021 | 0.0016 |
>
> In contrast, the test and train errors of EQGNN can decrease as the parameter counts increase, and the generalization errors are always small, such that EQGNN can achieve better results than GNN on the unfoldable dataset. In addition, we can compare the GNN with $d=1$ with EQGNN with $T=3$, because they have the same number of parameters. In this case, EQGNN can achieve better results than GNN in both training and testing. Similarly, comparing GNN with $d=2$ with EQGNN with $T=9$, we can obtain similar results. Furthermore, as we can see, only when GNN has more than 696 parameters does its train performance exceed EQGNN with 120 parameters. However, GNN with 696 parameters has poor generalization performance, such that the test performance of GNN is much lower than that of EQGNN. We have added the above to Appendix H in our revised paper.
>
> To further study the generalization performance of GNN, we test GNN on a larger dataset, i.e., the number of variables is 20, and the number of constraints is 6. The results of the GNN are:
> |                      |     |       |       |       |       |       |       |       |       |       |       |       |       |       |      |       |
> |:--------------------:| --- |:-----:|:-----:|:-----:|:-----:|:-----:|:-----:|:-----:|:-----:|:-----:|:-----:|:-----:|:-----:|:-----:|:----:|:-----:|
> |     **EmbSize**      | 1   |   2   |   3   |   4   |   5   |   6   |   7   |   8   |   9   |  10   |  11   |  12   |  16   |  32   |  64  |  128  |
> | **# Parameters**     | 60  | 180   | 360   | 600   | 900   | 1260  | 1680  | 2160  | 2700  | 3300  | 3960  | 4680  |  8160 | 31680 |124800| 495360|
> | **Train error rate** | 0.5 | 0.339 | 0.264 | 0.215 | 0.165 | 0.094 | 0.039 | 0.001 |   0   |   0   |   0   | 0.001 |   0   |   0   |  0   |   0   |
> | **Test error rate**  | 0.5 | 0.421 | 0.346 | 0.376 | 0.405 | 0.435 | 0.432 | 0.406 | 0.433 | 0.432 | 0.395 | 0.375 | 0.409 | 0.394 | 0.43 | 0.381 |
>
> From the result, we can see that the training error can decrease continuously, but the test error does not decrease starting from embedding size 3. In other words, when embedding size=3, it is the best GNN model on the test set, but the number of GNN parameters at this time is not enough to fit the training set well. As the number of parameters increases, GNN gets better at fitting the training set, but GNN does not generalize to the test set. This experiment shows the problem of GNN generalization on the MILP dataset. The above performance results of the GNN are obtained by using the code provided by the original paper (Chen et al., 2023b)(https://github.com/liujl11git/GNN-MILP).

---

> ### Author Response · Authors · 2023-11-19
> **Response to Reviewer 43Zw (Part 3)**
>
> >***W2: It is also unclear to me whether there is actually any theoretical quantum advantage when the quantum model has no auxiliary qubits as arXiv:2211.16998 gives efficient classical simulation algorithms for permutation-invariant quantum systems and machine learning models. This might limit the utility of the introduced quantum model to the case where there are many auxiliary qubits, which then runs into problematic training behavior as the authors point out.***
>
> **A2:** Thanks for your comment. One point we would like to clarify is that the contribution of our model may not lie in the quantum advantage you mentioned, i.e., the proposed quantum model cannot be simulated by efficient simulation algorithms. In other words, even if our model can be classically simulated, it still has an advantage. As stated by Reviewer pyPd, our proposed EQGNN can "overcome a fundamental limitation of traditional GNNs (i.e., GNNs can not distinguish pairs of *foldable MILP instances*)". More specifically, according to statistics, around 1/4 of the problems in MIPLIB 2017 (Gleixner et al., 2021) [1] involved *foldable MILPs*. It means that practitioners using GNNs cannot benefit from that if there are foldable MILPs in their dataset of interest (Chen et al., 2023b) [2]. In contrast, our proposed EQGNN is able to distinguish graphs that GNN cannot distinguish while maintaining permutation equivariance, and thus can be applied to general MILP instances, which is important and meaningful for the MILP community.
>
> Moreover, it may be good news if our model can be simulated by efficient classical algorithms. This is because classical GNNs have been adopted to represent mappings/strategies for MILP, for example, approximating Strong Branching (Gupta et al., 2020)[3], and parameterizing cutting strategies (Paulus et al., 2022)[4]. Based on the fundamental limitation of traditional GNNs, GNNs may struggle to predict a meaningful neural diving or branching strategy [1]. If our model can be simulated by efficient classical algorithms, we can replace the GNN part of these methods to better integrate with the classical framework. Thank you for letting us realize that our article is not clear enough in this part. We have added the above description to our revised paper.
>
> References:
>
> [1] Ambros Gleixner, Gregor Hendel, Gerald Gamrath, Tobias Achterberg, Michael Bastubbe, Timo Berthold, Philipp M. Christophel, Kati Jarck, Thorsten Koch, Jeff Linderoth, Marco Lubbecke, Hans D. Mittelmann, Derya Ozyurt, Ted K. Ralphs, Domenico Salvagnin, and Yuji Shinano. MIPLIB 2017: Data-Driven Compilation of the 6th Mixed-Integer Programming Library. Mathematical Programming Computation, 2021
>
> [2] Ziang Chen, Jialin Liu, Xinshang Wang, and Wotao Yin. On representing mixed-integer linear programs by graph neural networks. In Proceedings of the International Conference on Learning Representations, 2023b
>
> [3] Prateek Gupta, Maxime Gasse, Elias Khalil, Pawan Mudigonda, Andrea Lodi, and Yoshua Bengio. Hybrid models for learning to branch. Advances in neural information processing systems, 33: 18087–18097, 2020.
>
> [4] Max B Paulus, Giulia Zarpellon, Andreas Krause, Laurent Charlin, and Chris Maddison. Learning to cut by looking ahead: Cutting plane selection via imitation learning. In International Conference on Machine Learning, pp. 17584–17600. PMLR, 2022.
>
>
> >***W3: A much more minor point, but there are also many typos: "and yielding" at the bottom of page 3, "expressivity power" in the abstract, "guarantee to distinguish" in the abstract, and "TorchQauntum" at the bottom of page 7 are some that I found.***
>
> **A3:** Thanks for pointing out our grammatical/typographical mistakes.
> * "and yielding" should be modified to "yield" to keep the form consistent with the previous verb "learn".
> * "expressivity power" should be modified to "expressive power".
> * "can guarantee to distinguish" can be modified to "can distinguish" because 'guarantee' is not appropriate here.
> * "TorchQauntum" should be modified to "TorchQuantum".
> We have corrected these errors in the latest uploaded PDF, and further checked the full text for grammatical/typographical mistakes.

---

> ### Author Response · Authors · 2023-11-19
> **Response to Reviewer 43Zw (Part 4)**
>
> >***Q1: What are the connections between this work and previous work on permutation-invariant quantum machine learning models (cited above)?***
>
> **A4:** Thanks for your question. If we understand correctly, "(cited above)" refers to arXiv:2211.16998 you mentioned. Our proposed model is quite different from the mentioned work. The work (arXiv:2211.16998) focuses on calculating ground states and time-evolved expectation values for permutation-invariant Hamiltonians. Some simple combinatorial optimization problems, such as the Maxcut problem, can correspond to a  problem Hamiltonian, and we can find the solution by solving the ground state of the Hamiltonian. However, the way we solve the MILP problem is quite different from this class of problems. The MILP problem is abstracted as a bipartite graph, and predicting feasibility/optimal value/optimal solution is abstracted as a **supervised** graph-level/node-level classification/regression learning task. This can also be seen in the loss function used in section 3.5 of our latest submitted PDF.
>
>
> >***Q2: What explains the terrible generalization performance of the classical GNNs? The authors should perform additional numerics fixing the parameter counts of the classical and quantum models to control for this behavior.***
>
> **A5:** Thanks for your question. As reported in [Response to W1](), **(1)** the number of parameters of the GNN is limited by its structure and is related to the embedding sizes. We cannot guarantee the parameters of GNN are exactly the same as our EQGNN. But we can compare their performance on a similar number of parameters. For example, we find two sets of results from the table above.
>
> |                  | EQGNN($T=2$) | GNN($L=1,d=1$) | EQGNN($T=8$) | GNN($L=1,d=2$) |
> | ---------------- | ------------ | -------------- | ------------ | --- |
> | # Parameters     |   36     |        36    |     108      |  104  |
> | Train error rate | 0.0868        |   0.5          |  0.0632       |0.1216     |
> | Test error rate  | 0.0913       |  0.5          |   0.0655      |  0.1285 |
>
> As we can see, even though GNN has a similar number of parameters to our EQGNN, it still doesn't perform very well. **(2)** The table of [Response to W1](https://openreview.net/forum?id=KbvKjpqYQR&noteId=1B3XtxeDaW) shows that GNN does not have good generalization error bounds in the MILP dataset, resulting in poor generalization performance. In fact, MILP  is a very difficult problem, and Chen et al., 2023b are the first to apply GNN directly to approximate the solution of the MILP. The previous works only use GNN to help one part of their MILP algorithm, such as approximating Strong Branching or parameterizing cutting strategies. Moreover, Chen et al., 2023b only show how well GNN fits MILP training samples in their paper but do not show the performance of generalization on test sets. According to them,  the generalization for MILPs on the foldable dataset is good, but the generalization for MILPs on the unfoldable dataset is not good enough. This is indeed the case in our tests. Figure 6 in our latest PDF (Figure 8 of the original PDF) shows the generalization error of GNN is small in the foldable dataset, but the generalization error on the unfoldable dataset is not good, as shown in the above tables. This may be caused by the difference between the MILP test set and the training set. In contrast, the good generalization performance shown by the proposed EQGNN is able to benefit the MILP community.
>
> We hope that the above answers can address your concern. If you have any further questions, please contact me again. I am looking forward to your reply.

---

> > ### Comment · Reviewer_43Zw · 2023-11-21
> >
> > Thanks to the authors for following-up with clarifying numerics; they are much more convincing than the original numerics presented. I have upped my score accordingly.
> >
> > I in some sense agree that the presented architecture may still be interesting even if it is essentially a classical model by the results of arXiv:2211.16998. This ties into a question from Reviewer jrPx in that it is still unclear where the separation seen numerically is actually coming from---seemingly it is not from anything quantum mechanical, given the results of arXiv:2211.16998, which makes me think maybe this separation is just due to choosing a "bad" classical model to compare against. I will have to defer to the other Reviewers on this, though, since I am not an expert on state-of-the-art GNNs.

---

> > > ### Author Response · Authors · 2023-11-22
> > > **Second Response to Reviewer 43Zw (Part 1)**
> > >
> > > Dear Reviewer 43Zw,
> > >
> > > Thank you very much for your reply. We would like to express our sincere gratitude to you for raising our score. According to your reply, we infer that your concern is whether the performance/advantage of our model comes from the quantum mechanical. For this question, we give the following explanation, which shows how EQGNN uses *quantum properties* to achieve better separation power than GNN.
> > >
> > > As mentioned earlier, in the general MILPs, there is a class of MILPs called foldable MILPs, such as [Figure 2](https://postimg.cc/R64RKWvr) of our paper. GNN will *fail completely* on these *foldable* MILPs, as shown in the green line of [Figure 6](https://postimg.cc/23VQvdWq). Note that the mentioned foldable MILPs are different unfoldable MILPs, and the presented numerical experiments in the previous rebuttal only use the unfoldable MILPs for fairness, because GNNs are unable to deal with *foldable* MILPs. This inability is due to the *theoretical limitation* of GNNs based on the message-passing mechanism. Note that the GNNs based on the message-passing mechanism are a class of popular GNNs, including MPNN[1], vanilla GCN[2], GraphSage[3], MoNet[4], GAT[5] and GatedGCN[6]. GCN is used for our experiments on behalf of this class of GNNs. In other words, Figure 6 means that EQGNN is not limited to being superior to the GCN but surpasses a class of GNNs.  As pointed out by Xu et al. (2019) [7] and Morris et al. (2019) [8], GNNs based on message passing can never be more powerful than the 1-dimensional WeisfeilerLehman (1-WL) test in distinguishing non-isomorphic graphs. For example, [Figure 2](https://postimg.cc/R64RKWvr) is a pair of non-isomorphic MILP graphs (they have different edge connectivity and different feasibility), but GNN and 1-WL test cannot distinguish them, i.e., the algorithm will give the identical graph representation for these two MILP graphs. Next, we will explain why the 1-WL test and GNN cannot distinguish them and how EQGNN uses *quantum properties* to distinguish them.
> > >
> > > First, we present a variant of the WL test specially modified for MILP that follows the same lines as in (Chen et al. 2023b, Algorithm 1) [9].
> > >
> > > **Algorithm 2:** WL test for MILP-Graphs
> > >
> > > **Input:** A graph instance $(G, H) \in \mathcal{G}_{q,p} \times \mathcal{H}^V_q \times \mathcal{H}^S_p$, and iteration limit $L > 0$.
> > > 1. Initialize with $c_v^0 = hash(\boldsymbol f^V)$ for all $v\in V$, $c_s^0 = hash(\boldsymbol f^S)$ for all $s\in S$.
> > > 2. **for** $l = 1,2,...L$ **do**
> > > 3. $\quad c_{v_i}^l = hash(c_{v_i}^{l-1}, \sum_{j=0}^{p-1} A_{i,j} hash(c_{s_j}^{l-1}))$, for all $v \in V$
> > > 4. $\quad c_{s_j}^l = hash(c_{s_j}^{l-1}, \sum_{i=0}^{q-1} A_{i,j} hash(c_{v_i}^{l-1}))$, for all $s \in S$
> > > 4. **end for**
> > > 5. **return** the multisets containing all colors $\\{\\{c_v^l: v\in V, c_s^l: s\in S \\}\\}$.
> > >
> > > References:
> > >
> > > [1] Justin Gilmer, Samuel S Schoenholz, Patrick F Riley, Oriol Vinyals, and George E Dahl. Neural message passing for quantum chemistry. In International conference on machine learning, pp. 1263–1272. PMLR, 2017.
> > >
> > > [2] Thomas N. Kipf and Max Welling. Semi-supervised classification with graph convolutional networks. In International Conference on Learning Representations (ICLR), 2017
> > >
> > > [3] Will Hamilton, Zhitao Ying, and Jure Leskovec. Inductive representation learning on large graphs. In Advances in Neural Information Processing Systems, pages 1024–1034, 2017.
> > >
> > > [4] Federico Monti, Davide Boscaini, Jonathan Masci, Emanuele Rodola, Jan Svoboda, and Michael M. Bronstein. Geometric deep learning on graphs and manifolds using mixture model cnns. 2017 IEEE Conference on Computer Vision and Pattern Recognition (CVPR), Jul 2017.
> > >
> > > [5] Petar Velickovi, Guillem Cucurull, Arantxa Casanova, Adriana Romero, Pietro Liò, and Yoshua ´ Bengio. Graph Attention Networks. International Conference on Learning Representations, 2018
> > >
> > > [6] Xavier Bresson and Thomas Laurent. Residual gated graph convnets. arXiv preprint arXiv:1711.07553, 2017
> > >
> > > [7] Keyulu Xu, Weihua Hu, Jure Leskovec, and Stefanie Jegelka. How powerful are graph neural
> > > networks? In International Conference on Learning Representations, 2019.
> > >
> > > [8] Christopher Morris, Martin Ritzert, Matthias Fey, William L Hamilton, Jan Eric Lenssen, Gaurav Rattan, and Martin Grohe. Weisfeiler and leman go neural: Higher-order graph neural networks. In Proceedings of the AAAI conference on artificial intelligence, volume 33, pp. 4602–4609, 2019.
> > >
> > > [9] Ziang Chen, Jialin Liu, Xinshang Wang, and Wotao Yin. On representing mixed-integer linear programs by graph neural networks. In Proceedings of the International Conference on Learning Representations, 2023
> > >
> > > (To be Continued)

---

> > > ### Author Response · Authors · 2023-11-22
> > > **Second Response to Reviewer 43Zw (Part 3)**
> > >
> > > In addition, the 'GNN+rand' represents the proposed method by [9] to improve the results of GNN on the foldable dataset, which appends the vertex features of foldable MILPs with an additional random feature. However, adding random features is likely to change the feasibility or solution of the original problem, thus causing the ground truth of the dataset to change, so this treatment is still controversial.
> > >
> > > We hope the above can provide a better understanding of our paper. The proposed EQGNN can use the entanglement property of quantum to distinguish two MILP graphs that cannot be distinguished by the 1-WL test or a class of GNNs. From this perspective, we demonstrate the separation power of EQGNN can outperform traditional GNNs based on message passing, we believe that this can be regarded as the quantum advantage of our model.

---

> ### Author Response · Authors · 2023-11-22
> **Second Response to Reviewer 43Zw (Part 2)**
>
> The MILP graph is represented by $G = (V \cup S, A)$ and
> $\mathcal{G}_{q,p}$
> as the collection of all such weighted bipartite graphs whose two vertex groups (variables and constraints) have size $q$ and $p$, respectively. All the vertex features are stacked together as
> $H = (\boldsymbol{f}^V_1,...,\boldsymbol{f}^V_q,\boldsymbol{f}^S_1,...,\boldsymbol{f}^S_p) \in \mathcal{H}^V_q \times \mathcal{H}^S_p$.
>
> $hash(·)$ is a function that maps its input feature to a color in $\mathcal C$. The algorithm flow can be seen as follows. First, all nodes in $V \cup S$ are assigned to an initial color $c_v^0$ and $c_s^0$ according to their node features. Then, for each $v_i \in V$, $hash$ function maps the previous color
> $c_{v_i}^{l-1}$
> and aggregates the color of the neighbors of $\\{c_{s_j}^{l-1}\\}_{{s_j}\in \mathcal N(v_i)}$.
>
> Similarly, $hash$ function maps the previous color $c_{s_j}^{l-1}$ and aggregates the color of the neighbors of $\\{c_{v_i}^{l-1}\\}_{{v_i}\in \mathcal N(s_j)}$. This process is repeated until $L$ reaches the maximum iteration number. Finally, a histogram $h_G$ of the node colors can be obtained according to $\\{\\{c_v^l: v\in V, c_s^l: s\in S \\}\\}$, which can be used as a canonical graph representation. The notation $\\{\\{·\\}\\}$ denotes a multiset, which is a generalization of the concept of a set in which elements may appear multiple times: an unordered sequence of elements. That is, the 1-WL test transforms a graph into a canonical representation, and then if the canonical representation of two graphs is equal, the 1-WL test will consider them isomorphic.
>
> Next, we show a case where the 1-WL test fails. Taking [Figure 2](https://postimg.cc/R64RKWvr) in our paper as an example, we feed (a) and (b) from Figure 2 into this algorithm separately and compare the differences at each step. In the setting of Figure 2, $\{\boldsymbol{f}^V_i\}$ are equal, $\{\boldsymbol{f}^S_j\}$ are equal, are equal, and the weights of all edges are equal. In these two graphs, each node has two neighbors, but the only difference lies in their topological structure. More importantly, these two graphs have different feasibility, i.e., (a) is feasible while  (b) is infeasible. Initially, each variable vertex has the same color $c_v^0$, and each constraint vertex has the same color $c_s^0$, because their features are the same. This step is the same for Figure 2(a) and Figure 2 (b). Then, each vertex iteratively updates its color based on its own color and information from its neighbors. Note that, since each $v_i$ is connected to two $s_{j_1}$, $s_{j_2}$ in both Figure 2(a) and 2(b), they will also obtain the consistent $c_v^l$ and $c_s^l$ at this step. That is, Figure 2(a) and 2(b) will get the same representation by the algorithm, and they cannot be distinguished by 1-WL test GNNs. Moreover, Xu et al. [7] show that **GNNs are *at most* as powerful as the 1-WL test** in distinguishing graph structures. Therefore, a pair of 1-WL indistinguishable graphs are indistinguishable by GNNs.
>
> For EQGNN, the subfigure (a) and (b) of Figure 2 can be clearly distinguished. Since the two graphs differ only in the connectivity of edges, we only consider the modules associated with edges in EQGNN, i.e., $U_{g1}$ and $U_{g2}$. Specifically, $U_{g_1}(G,\beta) =  \text{exp}(-\textbf{i}(\sum_{(i,j)\in \mathcal E} (A_{i,j}+\beta) Z_{2i}Z_{2q+j}))$, where $i\in\\{0,1,...,q\\}$ indicates the index of variable nodes and $j\in\\{0,1,...,p\\}$ indicates the index of constraint nodes. Since two qubits represent one variable node, $2q$ qubits are used to represent all variables, and the index of the qubit representing the constraint node is numbered from $2q$. In subfigure (a) node $v_3$ is connected to $s_3$ and $s_4$, while in subfigure (b) node $v_3$ is connected to $s_3$ and $s_1$. Therefore, the difference between $U_{g_1}(G_a,\beta)$ and  $U_{g_1}(G_b,\beta)$ lies in the two-qubit gate acted on between qubits representing $v_3$ and **$s_4$** and the two-qubit gate acted on between qubits representing $v_3$ and $s_1$.  After the nodes correspond to their qubit index, we can obtain the  formulas $\text{exp}(-\textbf{i}((A_{3,4}+\beta) Z_{6}Z_{16}))$
> and $\text{exp}(-\textbf{i}((A_{3,1}+\beta) Z_{6}Z_{13}))$.
> Since the edge weights and the parameter $\beta$ are equal, we can just compare $\text{exp}(-\textbf{i}(Z_{6}Z_{13}))$ and $\text{exp}(-\textbf{i}(Z_{6}Z_{16}))$. These two values are obviously not equal, so $U_{g_1}(G_a,\beta) \neq U_{g_1}(G_b,\beta)$ and the overall unitary of our model $U(G_a,H) \neq U(G_b,H)$. Thereby, EQGNN can distinguish such graphs that the 1-WL test and GNNs cannot distinguish. We have added the above to the revised paper, see Appendix B. The experiments in Figure 6 show that the performance of EQGNN can achieve an error rate close to zero in these foldable datasets. In contrast, the error rate of GNNs stabilizes at 0.5.
>
> (To be Continued)

---

### Official Review · Reviewer_AMEi · 2023-10-25

**Soundness:** 3 good
**Presentation:** 3 good
**Contribution:** 2 fair
**Rating:** 5
**Confidence:** 4

**Summary:**

This paper proposes equivariant quantum graph neural networks (EQGNN) to solve mixed-integer linear programming (MILP) problems. In particular, their EQGNNs solve the issue of GNNs not being able to distinguish between so-called foldable instances; that is, MILP instances which have the same MILP-induced graph (up to permutations).
It is emphasized that their ansatz respects equivariance, since a permutation of the graph vertices results in the same permutation of the variables.
In addition, they conduct experiments which show good trainability and several other advantages over standard GNNs, including faster convergence, fewer parameters needed, and better generalization.

**Strengths:**

- In general, the presentation is fairly clear, and I can easily understand the overall motivations and contributions of this work.
- They are attempting to solve mixed-integer LP in an interesting and unique way. At the least, I have not seen MILPs solved this way.
- The experiments seem to be somewhat promising, showing benefits over GNNs.

**Weaknesses:**

- I feel that many parts of the construction of the ansatz is not well motivated. However, this seems to be a typical issue for quantum neural networks, perhaps even moreso than classical neural networks.
- Unless I am misunderstanding something, I feel that this permutation equivariance is not particularly insightful. For instance, the equivalent circuits in Figure 7 seem obvious, just that the circuit wires are drawn to either have different input order or output order. Perhaps it would be more useful to show a construction that fails to satisfy permutation equivariance.
- There are some details that I feel are important (at least for understanding) but left out (see Questions).

**Questions:**

Comments:
- In the definition of the feasible region $X_{fea}$ in Section 2, the constraint $l \leq x \leq u$ is missing.
- In Figure 6, I feel the circuit for $R_{zz}$ gates is somewhat misleading. The cross symbol is typically used for the SWAP gate. Also, the $R_{zz}$ is symmetric with respect to the two qubits it acts on, so there is no difference between choosing which is the target and which is the control qubit.
- Typo in the sentence "the identical parameteric gates are acted when the order of input nodes."
- Typo in the sentence "We now study the effect of width of the circuit increased"

Questions:
- It is not clear to me why there are instances of MILP that cannot be distinguished by GNNs, as in Figure 2. While the vertex degrees are the same, the connectivity is different, so shouldn't GNNs treat them differently? Perhaps I am missing something about how standard GNNs deal with this problem in the context of MILP.
- Why choose $R_{zz}$ over other two-qubit gates that commute with itself such as $R_{xx}$ or $R_{yy}$? This goes back to the weakness of the circuit construction not being well motivated.
- In the feature encoding layer, is there any reason why $R_x(c_i)$ and $R_x(u_i)$ as well as $R_x(l_i)$ and $R_x(\epsilon_i)$, are applied to different qubits?
- Out of curiosity, how the MILP instances are generated to be foldable or unfoldable? I suppose this requires something like solving the graph isomorphism problem.
- When predicting the solution vector, what is the actual representation of the classical information after reading out the qubits? For example, does a 2-qubit state for $v_1$ correspond to integers 0,1,2,3? Also, what happens when reading out a result that does not satisfy the constraints?

---

> ### Author Response · Authors · 2023-11-19
> **Response to Reviewer AMEi (Part 1)**
>
> Thanks for your insightful comments. We appreciate that some strengths of our paper are recognized, such as fairly clear presentation, and the interesting and unique way to solve MILPs. However, there seems to be some misunderstanding concerning our article, and we will do our best to provide a detailed explanation to address your concerns and improve our manuscript according to your suggestions. This is followed by point-to-point responses.
>
> >***Question 1: It is not clear to me why there are instances of MILP that cannot be distinguished by GNNs, as in Figure 2. While the vertex degrees are the same, the connectivity is different, so shouldn't GNNs treat them differently? Perhaps I am missing something about how standard GNNs deal with this problem in the context of MILP.***
>
> **A1:** This inability to distinguish is related to **the essential limitation of the neighbor-aggregation mechanism** of GNNs, and this limitation does not only exist in the context of MILP, which we will elaborate on next.
>
> To better understand this limitation, let's first take a more fundamental example, which is also not distinguishable by GNNs. The problem of distinguishing two graphs can be abstracted as a graph isomorphism testing problem, i.e., to determine whether two graphs are isomorphic.
>
> *Definition 1: Two graphs G1 and G2 are isomorphic if there exists a matching between their vertices so that two vertices are connected by an edge in G1 if and only if corresponding vertices are connected by an edge in G2.*
>
> That is, two graphs that are isomorphic means that they will have the same topological structure or connectivity. Let's look at a simple example of non-isomorphic graphs, which you can see by clicking on this [Figure R1](https://postimg.cc/56YC7GMt). Both graphs are *2-regular graphs*, and the regular graph is a graph where each vertex has the same number of neighbors, i.e., every vertex has the same degree. The only difference is that Figure R1 (a) has two cycles, and Figure R1 (b) has only one cycle.
>
> The 1-dimensional Weisfeiler-Lehman (1-WL) algorithm is a well-known heuristic for deciding whether two graphs are isomorphic. It is known to work well in general, **with a few exceptions, e.g., regular graphs** [1,2]. Moreover, Xu et al. [3] show that **GNNs are *at most* as powerful as the 1-WL test** in distinguishing graph structures. Therefore, a pair of 1-WL indistinguishable graphs are indistinguishable by GNNs. Next, we will briefly introduce the 1-WL test algorithm and why the [Figure R1](https://postimg.cc/56YC7GMt) cannot be distinguished by 1-WL or by GNNs.
>
> **Algorithm 1:** 1-WL (color refinement)
>
> **Input:** $G = (V, E, X_V)$
> 1. $c_v^0 = hash(X_v)$ for all $v\in V$
> 2. **repeat**
> 3. $\quad c_v^l = hash(c_v^{l-1}, \\{\\{c_w^{l-1}: w \in  \mathcal N_G(v)\\}\\})$, for all $v \in V$
> 4. **until** $c_v^l= c_v^{l-1}$ for all ${v\\in V}$
> 5. **return** $\\{\\{c_v^l: v\in V \\}\\}$
>
> In the above algorithm, we consider an undirected graph $G = (V,E,X_V)$, where $V$ is the set of vertices; $E$ is the set of edges; and $X_V$ is the set of node features. The notation $\\{\\{·\\}\\}$ denotes a multiset, which is a generalization of the concept of a set in which elements may appear multiple times: an unordered sequence of elements. $hash(·)$ is a function that maps its input to a color in $\mathcal C$. The algorithm flow can be seen as follows. First, all nodes in $G$ are assigned to an initial color $c_v^0$ according to their node features. Then, $hash$ function maps $v$’s previous color $c_v^{l-1}$ and the multiset of the previous colors of $v$’s neighbors $\\{\\{c_w^{l-1}: w \in  \mathcal N_G(v)\\}\\}$ to a color $c_v^l$ in $\mathcal C$, for each $v \in V$. This process is repeated until the colors of all nodes do not change, i.e., $c_v^l= c_v^{l-1}$ for all ${v\\in V}$. Finally, a histogram $h_G$ of the node colors can be obtained according to $\\{\\{c_v^l: v\in V \\}\\}$, which can be used as a canonical graph representation. That is, the 1-WL test transforms a graph into a canonical representation, and then if the canonical representation of two graphs is equal, 1-WL will consider them isomorphic.
>
> In [Figure R1](https://postimg.cc/56YC7GMt), we assume that all node features are equal, and thereby all nodes are assigned to the same initial color. Then, the hash function aggregates the color of $v$ and the colors of the two neighbors of $v$. It should be noted that Figure R1 (a) and Figure R1 (b) have the same result in this step because their vertex degrees and node features are the same. This also directly leads to the fact that Figure R1 (a) and Figure R1 (b) will certainly have the same canonical graph representation after the 1-WL algorithm. However, they are clearly non-isomorphic graphs, and 1-WL fails to distinguish them.
>
> (to be continued)

---

> ### Author Response · Authors · 2023-11-19
> **Response to Reviewer AMEi (Part 2)**
>
> As mentioned earlier, one well-known limitation of GNNs is their separation power, which is upper bounded by the 1-WL algorithm for graph isomorphism testing [3] because they have similar neighbor aggregation mechanisms. Therefore, for a pair of 1-WL indistinguishable graphs, as shown in Figure R1 or Figure 2, any GNNs will learn the exact same representations for these graphs, yielding the same prediction for both.
>
>
> As we can see, Figure 2 differs from [Figure R1](https://postimg.cc/56YC7GMt) in that it is a bipartite graph for the MILP task. To prove that Figure 2 is also a pair of 1-WL indistinguishable graphs, we present a variant of 1-WL test specially modified for MILP that follows the same lines as in (Chen et al. 2023b, Algorithm 1)[4].
>
> **Algorithm 2:** WL test for MILP-Graphs
>
> **Input:** A graph instance $(G, H) \in \mathcal{G}_{q,p} \times \mathcal{H}^V_q \times \mathcal{H}^S_p$, and iteration limit $L > 0$.
> 1. Initialize with $c_v^0 = hash(\boldsymbol f^V)$ for all $v\in V$, $c_s^0 = hash(\boldsymbol f^S)$ for all $s\in S$.
> 2. **for** $l = 1,2,...L$ **do**
> 3. $\quad c_{v_i}^l = hash(c_{v_i}^{l-1}, \sum_{j=0}^{p-1} A_{i,j} hash(c_{s_j}^{l-1}))$, for all $v \in V$
> 4. $\quad c_{s_j}^l = hash(c_{s_j}^{l-1}, \sum_{i=0}^{q-1} A_{i,j} hash(c_{v_i}^{l-1}))$, for all $s \in S$
> 4. **end for**
> 5. **return** the multisets containing all colors $\\{\\{c_v^l: v\in V, c_s^l: s\in S \\}\\}$.
>
> We can represent the MILP graph as  $G = (V \cup S, A)$
> and $\mathcal{G}_{q,p}$
> as the collection of all such weighted bipartite graphs whose two vertex groups have size $q$ and $p$, respectively. All the vertex features are stacked together as $H = (\boldsymbol{f}^V_1,...,\boldsymbol{f}^V_q,\boldsymbol{f}^S_1,...,\boldsymbol{f}^S_p) \in \mathcal{H}^V_q \times \mathcal{H}^S_p$. Thereby, the weighted bipartite graph with vertex features
>
> $(G,H) \in \mathcal{G}_{q,p} \times \mathcal{H}^V_q \times \mathcal{H}^S_p$
>
> contains all information on the MILP problem. Next, we feed (a) and (b) from Figure 2 into this algorithm separately and compare the differences at each step. In the setting of Figure 2, $\{\boldsymbol{f}^V_i\}$ are equal,
> $\{\boldsymbol{f}^S_j\}$ are equal, and the weights of all edges are equal. Therefore, initially, each variable vertex has the same color $c_v^0$, and each constraint vertex has the same color $c_s^0$. This step is the same for Figure 2(a) and Figure 2 (b). Then, each vertex iteratively updates its color, based on its own color and information from its neighbors. Note that, since each $v_i$ is connected to two $s_{j_1}$, $s_{j_2}$ in both Figure 2(a) and 2(a), they will also obtain the consistent $c_v^l$ and $c_s^l$ at this step. That is, Figure 2(a) and 2(b) will get the same representation by Algorithm 2, and they cannot be distinguished by 1-WL test and GNNs. We have added the above to [our latest uploaded PDF](https://openreview.net/pdf?id=KbvKjpqYQR), please see Appendix B.
>
> **References:**
>
> [1] Jin-Yi Cai, Martin Fürer, and Neil Immerman. An optimal lower bound on the number of variables for graph identification. Combinatorica, 12(4):389–410, 1992.
>
> [2] Brendan L Douglas. The weisfeiler-lehman method and graph isomorphism testing. arXiv preprint arXiv:1101.5211, 2011.
>
> [3] Xu, K., Hu, W., Leskovec, J., and Jegelka, S. How powerful are graph neural networks? In ICLR, 2019.
>
> [4] Ziang Chen, Jialin Liu, Xinshang Wang, and Wotao Yin. On representing mixed-integer linear programs by graph neural networks. In Proceedings of the International Conference on Learning Representations, 2023b.
>
>
>
> >***Question 2: Why choose $R_{zz}$ over other two-qubit gates that commute with itself such as $R_{xx}$ or $R_{yy}$? This goes back to the weakness of the circuit construction not being well motivated.***
>
> **A2:** First of all, we need to clarify that the *commute* in this article does not refer to $R_{zz}(\theta_{12})R_{zz}(\theta_{23}) = R_{zz}(\theta_{23})R_{zz}(\theta_{12})$, but to $[R_{zz}(\theta_{12})\otimes I][I\otimes R_{zz}(\theta_{23})] = [I\otimes R_{zz}(\theta_{23})][R_{zz}(\theta_{12})\otimes I]$. More specifically, it refers to $\text{exp}(-\text{i}(\theta_{i_1,j_1} Z_{i_1} Z_{j_1} + \theta_{i_2,j_2} Z_{i_2} Z_{j_2})) = \text{exp}(-\text{i}(\theta_{i_2,j_2} Z_{i_2} Z_{j_2} + \theta_{i_1,j_1} Z_{i_1} Z_{j_1}))$, where $Z_{i_1} = I \otimes ... Z_{i_1} \otimes ... \otimes I$. Since $\text{exp}(-\text{i} A_{i,j} Z_i Z_j)$ is a diagonal matrix, and all diagonal matrices are commute, the equation is proven. However, other two-qubit gates, \emph{e.g.}, $\text{exp}(-\textbf{i} A_{i,j} X_i X_j)$ does not satisfy the equation. In other words, the $\text{R}_\text{zz}(\theta) = \text{exp}(-\textbf{i} \theta Z_i Z_j)$ gate is carefully chosen. In addition, we have added the above to section 3.6 of the updated PDF and shown why this equation is required to hold.

---

> ### Author Response · Authors · 2023-11-19
> **Response to Reviewer AMEi (Part 3)**
>
> For a more intuitive understanding, let us give a simple example of why such a property is needed in our circuit and why $R_{zz}$ are chosen over other two-qubit gates. As shown in [Figure R2](https://postimg.cc/RWMVZVvD) (this is a hyperconnection), suppose there are three qubits representing three nodes, and there is an edge $e_{12}$ between $v_1$ and $v_2$, and there is an edge $e_{23}$ between $v_2$ and $v_3$. Each edge is associated with a two-qubit gate. For learning graphs, the model needs to ensure that the circuit is the same unitary regardless of the arrangement of the edges. For example, the order in which the two-qubit gate associated with the edges acts should not affect the model results. **$R_{zz}$ gate are carefully selected to ensure that the edge arrangement invariance, while $R_{xx}$ and $R_{yy}$ do not have this property.** For example, $R_{zz}$ is a diagonal matrix, then $I \otimes R_{zz}(\theta_{23})$ or $R_{zz}(\theta_{12}) \otimes I$ are still diagonal matrix, and that diagonal matrices commute, i.e., $[R_{zz}(\theta_{12})\otimes I][I\otimes R_{zz}(\theta_{23})] = [I\otimes R_{zz}(\theta_{23})][R_{zz}(\theta_{12})\otimes I]$. Hence, we can arbitrarily choose the order of action of the two-qubit gates associated with the edge. However, $[R_{xx}(\theta_{12})\otimes I][I\otimes R_{xx}(\theta_{23})] \neq [I\otimes R_{xx}(\theta_{23})][R_{xx}(\theta_{12})\otimes I]$ and $[R_{yy}(\theta_{12})\otimes I][I\otimes R_{yy}(\theta_{23})] \neq [I\otimes R_{yy}(\theta_{23})][R_{yy}(\theta_{12})\otimes I]$, which they do not meet the design requirements of QGNNs. Thank you for letting us realize that our article is not clear enough in this part. We have added the above description to our revised paper, please refer to our latest PDF submission, especially Theorem 3.
>
>
> >***Question 3: In the feature encoding layer, is there any reason why $R_x(c_i)$ and $R_x(u_i)$ as well as $R_x(l_i)$ and $R_x(\epsilon_i)$, are applied to different qubits?***
>
> **A3:** As mentioned in Section 3.2, we set a hyperparameter $\omega$ to control the number of features on each qubit in our method. Figure 5 of our paper only shows an example of $w = 2$. Each variable has four features, and two qubits represent it. Of course, we can also set $\omega$ to 4 so that $(c_i, u_i, l_i, \epsilon_i)$ is encoded on the same qubit. As we can see, the smaller the value of $w$, the larger the number of qubits representing one node. This means that the higher the dimension of the feature representation we can learn for each node, the more expressive the model will be. The choice of $\omega$ is a trade-off between the available quantum resources (the number of qubits) and the expressive power of the model. In addition, since each feature dimension is associated with a learnable parameter, the order in which these four features are used is not important.
>
> >***Question 4: Out of curiosity, how the MILP instances are generated to be foldable or unfoldable? I suppose this requires something like solving the graph isomorphism problem.***
>
> **A4:** Thanks for your questions. The generation of foldable or unfoldable MILP instances is consistent with that of the reference (Chen et al., ICLR 2023b) [1], which does not involve solving the graph isomorphism problem. Next, we will introduce the generation process in detail and add this part to our revised paper.
>
> Note that the foldable and unfoldable datasets are generated separately. *Foldable* dataset is constructed by many pairs of 1-WL *indistinguishable* graphs, and figure 2 in our paper is a foldable example, which is a pair of non-isomorphic graphs that cannot be distinguished by WL-test or by GNNs, as mentioned in the response to Question 1. In contrast, the *unfoldable* dataset refers to other MILP instances that do not have 1-WL indistinguishable graphs.
>
> In section 4.1, we randomly generate 2000 *foldable* MILPs with 12 variables and $6$ constraints, and there are 1000 feasible MILPs with attachable optimal solutions while the others are infeasible.
> We construct the $(2k − 1)$-th and $2k$-th problems via following approach $(1 ≤ k ≤ 500)$.
> * Sample $J = \\{j_1, j_2, . . . , j_6\\}$ as a random subset of $\\{1, 2, . . . , 12\\}$ with 6 elements.
> 1. For $j \in J$, $x_j \in \\{0, 1\\}$, i.e., $x_j$ is a binary integer variable.
> 2. For $j \notin J$, $x_j$ is a continuous variable with bounds $l_j∼ \mathcal{U}(0, \pi), u_j∼ \mathcal{U}(0, \pi)$. If $l_j > u_j$, then switch $l_j$ and $u_j$.
> * $c_1 =. . . = c_{12} = 0.01$.
> * The constraints for the $(2k−1)$-th problem (feasible) is $x_{j_1}+x_{j_2}=1$, $x_{j_2}+x_{j_3}=1$, $x_{j_3}+x_{j_4}=1$, $x_{j_4} + x_{j_5} = 1$, $x_{j_5} + x_{j_6} = 1$, $x_{j_6} + x_{j_1} = 1$. For example, $x = (0, 1, 0, 1, 0, 1)$ is a feasible solution.
>
> (to be continued)

---

> ### Author Response · Authors · 2023-11-19
> **Response to Reviewer AMEi (Part 4)**
>
> * The constraints for the $2k$-th problem (infeasible) is $x_{j_1}+x_{j_2}=1$, $x_{j_2}+x_{j_3}=1$, $x_{j_3}+x_{j_1}=1$, $x_{j_4} + x_{j_5} = 1$, $x_{j_5} + x_{j_6} = 1$, $x_{j_6} + x_{j_4} = 1$. An explanation for why it is infeasible: we add the first three formulas together to get $2(x_{j_1}+x_{j_2}+x_{j_3})=3$, but $x_j \in \\{0, 1\\\}$ for $j \in J$, so there will be no solution that satisfies this equation and the MILP problem must be infeasible.
>
> For unfoldable MILPs, we first set the number of variables and constraints to $m$ and $n$.
> * For each variable, $c_j ∼ \mathcal N(0, 0.01)$, $l_j, u_j ∼ \mathcal N(0, \pi)$. If $l_j > u_j$, then switch $l_j$ and $u_j$. The probability that $x_j$ is an integer variable is 0.5.
> * For each constraint, $\circ_i ∼ \mathcal U({≤, =, ≥})$ and $b_i ∼ \mathcal U(-1, 1)$.
> * After randomly generating all the MILP samples, we use the 1-WL test algorithm to calculate their graph representation for each instance, ensuring that there are no duplicate graph representations in the dataset so that we can determine that this dataset does not contain 1-WL indistinguishable pairs of MILP instances.
>
> We have added the above description of dataset generation to our revised paper, and please see Appendix D in the latest PDF submission. Thanks for your suggestion.
>
> References:
>
> [1] Ziang Chen, Jialin Liu, Xinshang Wang, and Wotao Yin. On representing mixed-integer linear programs by graph neural networks. In Proceedings of the International Conference on Learning Representations, 2023b.
>
> >***Question 5: When predicting the solution vector, what is the actual representation of the classical information after reading out the qubits? For example, does a 2-qubit state for $v_1$ correspond to integers 0,1,2,3? Also, what happens when reading out a result that does not satisfy the constraints?***
>
> **A5:** Thanks for your question. As mentioned earlier, the hyperparameter $\omega$ can control the number of qubits representing each node. If the number of qubits is more than 1, like the node $v_1$ in Figure 6, we will add the control gate at the end of the node update layer, such as the CRY gate in Figure 6. Then, at the end of the circuit, we only measure the first qubit representing the node. As mentioned in section 3.5 of our paper, we perform the Pauli-Z measurement on the qubit, which can obtain the $P(|0\rangle)-P(|1\rangle)$, which ranges from -1 to 1. That is, each variable node can obtain one value $F_{sol}^i(G)$ to predict the values of variables. In addition, the maximum range $\lambda$ of variables of the training sample is known. The solution vector of the final output of the model is
> $\\{\lambda F_{sol}^i(G)\\}_{i=0}^{q-1}$.
>
> Assuming the groundtruth of the solution vector is $\\{y_{sol}^i(G)\\}_{i=0}^{q-1}$, we evaluate the performance of the predicted solution vector by:
>
> $$\frac{1}{Mq}\sum_{m=0}^{M-1} \sum_{i=0}^{q-1}(y_{sol}^i(G_m)-\lambda F_{sol}^i(G_m))^2$$
>
> where $G_m$ indicates the $m$-th tested MILP graph, and the number of tested instances is $M$. As we can see, the values of the solution vector are continuous. For integer variables, a post-processing can be applied to make a rounding approximation. In addition, we need to make it clear that the proposed method is similar to the previous methods [1], only providing an approximate solution rather than providing an exact solution. We have added the above explanation to our latest uploaded PDF, and please see our highlights in Section 3.5.
>
> >***Comment 1: In the definition of the feasible region $X_{fea}$ in Section 2, the constraint $l \leq x \leq u$ is missing.***
>
> **A6:** Thank you for pointing this out, and we have corrected the definition of the feasible region $X_{fea}$ in our latest uploaded PDF.
>
> >***Comment 2: In Figure 6, I feel the circuit for $R_{zz}$ gates is somewhat misleading. The cross symbol is typically used for the SWAP gate. Also, the  $R_{zz}$ is symmetric with respect to the two qubits it acts on, so there is no difference between choosing which is the target and which is the control qubit.***
>
> (to be continued)

---

> ### Author Response · Authors · 2023-11-19
> **Response to Reviewer AMEi (Part 5)**
>
> **A7:** Thanks for your comment. The $R_{zz}$ gates with different control qubits indeed have the same unitary matrix. However, [Figure R3](https://postimg.cc/R6QKYKb3) shows an example of using different control qubits. As we can see, although the input quantum state and output quantum state of the whole circuit are kept consistent for Figure (a) and (b), the position where the global phase appears on a single qubit will be different, such as Figure (a) outputs $|\psi_0^a\rangle = |1\rangle$ and Figure (b) outputs $|\psi_0^b\rangle = e^{i\frac{\theta}{2}}|1\rangle$. However, the global phase $e^{i\frac{\theta}{2}}$ has no effect on the observations resulting from a measurement of the state, and it is only the artefact of the mathematical framework and has no physical meaning. We specify the control qubits solely for the certainty of the mathematical formulation. Nevertheless, our specification of control qubits and the use of cross symbols may cause misunderstanding, so we modified Figure 6 in our paper, please see Figure 5 of the latest submitted PDF. Thank you for your suggestion.
>
>
> >***Comment 3: Typo in the sentence "the identical parametric gates are acted when the order of input nodes."***
>
> **A8:** Thanks for pointing out the problem. The sentence is incomplete and should be modified to "the identical parametric gates are acted on the circuit no matter how the order of input nodes changes."
>
> >***Comment 4: Typo in the sentence "We now study the effect of width of the circuit increased"***
>
> **A8:** We apologize for not writing this sentence clearly. It should be modified to "We now study the influence on the performance with the increase of circuit width".
>
>
> >***Weakness 2: Unless I am misunderstanding something, I feel that this permutation equivariance is not particularly insightful. For instance, the equivalent circuits in Figure 7 seem obvious, just that the circuit wires are drawn to either have different input order or output order. Perhaps it would be more useful to show a construction that fails to satisfy permutation equivariance.***
>
> **A9:** First of all, we need to clarify that Figure 7 is just a very simplified example of showing permutation equivariance, and it does not represent our circuit, and designing permutation-equivariant circuits is not easy. To address your concerns, we have rewritten Section 3.5 and 3.6 in the latest uploaded PDF. Section 3.5 formulates the proposed model. In particular, **Section 3.6 presents four definitions and three theorems to prove the permutation equivariance of our proposed model**. Since this part is about two pages long, please refer to the highlighted section of our latest uploaded PDF. In addition, we also provide a shorter version to answer the question.
>
> One of the most intuitive examples of failure to satisfy permutation equivariance is HEA (Hardware efficient ansatz). [Figure R4](https://postimg.cc/qhCCzp3P) shows an example of an HEA with three qubits, which is used to learn the representation of graphs with three nodes. [Figure R4](https://postimg.cc/qhCCzp3P) uses $R_x$ gate to encode the feature of three nodes, and uses $RY(\theta_0), RY(\theta_1)$, $RY(\theta_2)$ and CNOT gate as the learning module. Suppose that the order of node is $(v_0, v_1, v_2)$, we measure each qubit and get the probability of $|0\rangle$ at each qubit as the output $(y_{v_0}, y_{v_1}, y_{v_2})$. Then, the order of node changes to $(v_1, v_0, v_2)$ and the output is
> $(y_{v_1}^{\\prime}, y_{v_0}^{\\prime}, y_{v_2}^{\\prime})$.
> Obviously, $y_{v_0} \neq y'_{v_0}$. The permutation equivariance of the model is vital for graph learning, and the output of the model should be reordered consistently with the permutation on the input nodes, otherwise the model may overfit to the order of nodes in the training data. In the field of classic machine learning, GNN can use the neighborhood-aggregation mechanism to achieve the permutation equivariance. However, in the field of quantum computing, the permutation-equivariant model requires fine designs due to the constraints of quantum systems.
>
> >***Weakness 1: I feel that many parts of the construction of the ansatz is not well motivated. However, this seems to be a typical issue for quantum neural networks, perhaps even moreso than classical neural networks.***
>
> **A10:** Thanks for your comment. In fact, because our network is required to satisfy permutation equivariance, each layer of our model must be designed for encoding or learning while ensuring permutation equivariance, which is much more motivated than other QNNs. We believe we have addressed some of your concerns in our previous answers **A2**, **A3**, and **A9**. Moreover, the new content added in section 3.6 in our current submitted PDF can further address your concerns.
>
> We hope that the above answers can address your concern. If you have any further questions, please contact me again. I am looking forward to your reply.

---

> > ### Comment · Reviewer_AMEi · 2023-11-21
> >
> > Thanks for the detailed explanations. Many details are much more clear to me, thanks to the responses and updated PDF document.
> >
> > I am still confused about the explanation regarding the choice of $ZZ$ over $XX$ or $YY$. For a simple 3 qubit case, $X\otimes X \otimes I$ obviously commutes with $I \otimes X \otimes X$, and similarly the same holds when replacing all $X$ operators with $Y$.

---

> ### Author Response · Authors · 2023-11-21
> **Response to Reviewer AMEi**
>
> Dear Reviewer AMEi,
>
> Thank you very much for your reply. We would like to express our sincere gratitude for your valuable feedback. You are right. It is true that $X \otimes X \otimes I$ commute $I \otimes X \otimes X$. We apologize that the previous statement in the rebuttal is inappropriate. $R_{XX}$ gate and $R_{YY}$ gate are similar to  $R_{ZZ}$ gate, and they can be used to preserve the permutation invariant of edges.  $R_{ZZ}$ gate is not the only option to preserve the permutation invariant of edges, and the two-qubit gates that can commute in the circuit are able to be used to learn the graph information interaction. We have added the above to our latest uploaded PDF, please see the second to last paragraph of section 3.6.
>
> In addition, we want to point out that our proposed model provides a general framework for designing equivariant QNN on learning graph structure data besides the specific construction of the circuits. Specifically, as mentioned in section 3.6, an equivariant GQNN can be built by the layers independent of input orders and the layers related to input orders. The former can be constructed by the proposed parameter-sharing mechanism and the latter can be established by choosing the two-qubit gates that can commute in the circuit to preserve the permutation of edges. Moreover, the proposed multi-qubit encoding mechanism and optional auxiliary layer can also provide new ideas for the construction of other EQNNs to handle high-dimensional data features. In other words, we provide a design rule to build an equivariant graph QNN.
>
> We are grateful for your feedback that helped improve our submission.
>
>
>
> Best regards

---

> ### Author Response · Authors · 2023-11-22
> **Thanks for your reply & We look forward to your further feedback**
>
> Dear Reviewer AMEi,
>
> Thanks again for your review. We have replied to the problem about the choice of two-qubit gates in [the last response](https://openreview.net/forum?id=KbvKjpqYQR&noteId=R27tE9Usid) (we have further updated it just now). We wanted to reach out to see if our response has addressed your concerns.
>
> In addition, we would appreciate it if you could consider updating our score.
>
> We are more than happy to discuss further if you have any further concerns and issues, please kindly let us know your feedback.
>
> May you have a blessed and happy Thanksgiving!
>
>
> With gratitude

---

> > ### Comment · Reviewer_AMEi · 2023-11-22
> >
> > Thanks for addressing some of my concerns. I have raised my score (3 to 5). I still feel the motivation behind the ansatz is not very principled and there should be some better study of how it is designed, down to the gate level.

---

### Official Review · Reviewer_jrPx · 2023-10-29

**Soundness:** 3 good
**Presentation:** 3 good
**Contribution:** 3 good
**Rating:** 5
**Confidence:** 3

**Summary:**

This submission presents Equivariant Quantum Graph Neural Network (EQGNN), a Variational Quantum Circuit (VQC), as a parametric model for data-driven Mixed Integer Linear Programming (MILP) solutions. The weighted bipartitle graph for MILP contains two types of nodes for decision variables and constraints, with the edges only connecting nodes of different types. Such a constructed graph is used as the input for VQC to predict the feasibility, optimal value and solution for MILP. The authors show that the feasibility, optimal value and solution to MILP are either equivariant or invariant to the permutation on the order of nodes and edges of the graph, and propose a VQC to encode the information into quantum states, which satisfy the equvariant or invariant properties of the weighted bipartitle graph representation of MILP. The main idea is the diagonality of $R_{zz}(\theta)$ gate and to use a shared set of parameters for nodes of the same type. Experiments show the potential capability of proposed model to solve MILP problems.

**Strengths:**

1. Carefully chosen gates for feature encoding and massage passing in EQGNN help achieve permutation equivariance or invariance, with the corresponding reasoning.

2. Experiments were performed to demonstrate the potential of EQGNN for MILP.

**Weaknesses:**

1. In Section 1, the authors claimed that "We propose a so-called (for the first time) Equivariant Quantum Graph Neural Network (EQGNN) capable of representing general MILP instances without the need for dataset partitioning." The authors need to clearly explain what it means by ``without the need for dataset partitioning'' and why it is important.

2. Also in Section 1, the authors claimed that "We both theoretically and empirically show the separation power of EQGNN can surpass that of GNNs in terms of representing MILP graphs." However, the current submission does not contain any theoretical analysis of the separation power of EQGNN.

3. In Section 4.1, it is unclear why and how the number of parameters are compared between neural networks in classic and quantum computers. The authors need to clarify how the criteria are chosen. In Section 4.2, it is also unclear why neural networks in classical computers with 2,096 and 7,904 parameters are chosen as the baseline. It will be better if the authors can compare the proposed model with a non-data-driven method for solving MILP.

4. Throughout the paper, there are many typos/errors/inconsistencies. For example, at the beginning of the third paragraph of Section 2: "Foladable MILP instances"; in equation (4) the authors use $R_x$ gate for encoding $l$ and $\epsilon$ but in Figure 5 $R_z$ is used to encode $l$ and $\epsilon$. It would be better to have consistent annotations for example in Section 2, the same $q$ decision variables and $p$ constraints can be used to index $V$ and $S$ in bipartite graph representation. There are many of these problems.

**Questions:**

1. Is that possible for two MILP problems with different feasible/optimal solutions to be encoded into the same quantum state with the same VQC parameters?

2. Node permutation equivariance or invariance comes naturally with graph representations. The authors need to clearly state whether the claimed generalizability, expressiveness, and efficiency should be attributed to EQGNN or VQC? What are the main technical contributions? How shall they be positioned in the context of Table 1? Also, why the mentioned reference Schatzki et al. (2022) was not included in Table 1?

3. The author should clearly describe how the VQC learning problems are formulated for MILP feasibility/optimal solutions/optimal values based on the proposed EQGNN respectively either in the main text or appendix. The author should also clearly specify how the binary measurements can be used to approximate the optimal value which is often a non-discrete value. If multiple measurements are performed to recover the approximated quantum state of the VQC, the author should also clearly specify how many measurements are needed for the results provided in experiment sections.

4. How expressive is the proposed model?

---

> ### Author Response · Authors · 2023-11-19
> **Response to Reviewer jrPx (Part 1)**
>
> >***Weakness 1: In Section 1, the authors claimed that "We propose a so-called (for the first time) Equivariant Quantum Graph Neural Network (EQGNN) capable of representing general MILP instances without the need for dataset partitioning." The authors need to clearly explain what it means by ``without the need for dataset partitioning'' and why it is important.***
>
> **A1:** Thanks for your comment. This sentence appears after introducing that the classic GNNs need to divide the whole MILP dataset into the *foldable* and *unfoldable* datasets. For unfoldable MILPs, GNNs can solve them directly. However, for foldable MILPs, GNNs require additional processing on the dataset to solve them, such as adding random features. This is because foldable MILPs contain many pairs of graphs that cannot be distinguished by GNNs, such as Figure 2, which has different feasibility for the MILP problem, but their graph representations learned by GNNs are the same due to the fundamental limitations of GNNs. The theoretical explanation of why GNNs cannot distinguish these two graphs can refer to our *response to Weakness 2*, and we have added it to our [lastest submitted PDF](https://openreview.net/pdf?id=KbvKjpqYQR). *More importantly, according to statistics, about 1/4 of the problems in MIPLIB 2017 (Gleixner et al., 2021) [1] involved foldable MILPs*. It means that practitioners using GNNs cannot benefit from that if there are foldable MILPs in their dataset of interest (Chen et al., 2023b) [2]. To this end, Chen et al. [2] proposed a method by appending the vertex features of foldable MILPs with an additional random feature. However, adding random features is likely to change the feasibility or solution of the original problem, thus causing the ground truth of the dataset to change, so this treatment is still controversial. In contrast, as stated by Reviewer pyPd, our proposed EQGNN can "overcome a fundamental limitation of traditional GNNs (i.e., GNNs can not distinguish pairs of foldable MILP instances)". In other words, the meaning of "EQGNN without the need for dataset partitioning" is that our proposed EQGNN can be applied to any MILP instance, and there is no need to make any differential treatment for all MILP instances. It is worth noting that this is one of the core contributions of our paper, which demonstrates the quantum advantage over classical graph neural networks in dealing with MILP problems. We apologize for the clarification of this point not being clear enough in our paper, so we have added more explanation about this sentence. Please refer to the highlighted section in our latest PDF submission. Thanks for your suggestion.
>
> References:
>
> [1] Ambros Gleixner, Gregor Hendel, et al. MIPLIB 2017: Data-Driven Compilation of the 6th Mixed-Integer Programming Library. Mathematical Programming Computation, 2021
>
> [2] Ziang Chen, Jialin Liu, Xinshang Wang, and Wotao Yin. On representing mixed-integer linear programs by graph neural networks. In Proceedings of the International Conference on Learning Representations, 2023b
>
> >***Weakness 2: Also in Section 1, the authors claimed that "We both theoretically and empirically show the separation power of EQGNN can surpass that of GNNs in terms of representing MILP graphs." However, the current submission does not contain any theoretical analysis of the separation power of EQGNN.***
>
> **A2:** Thanks for your suggestions. We have added the theoretical analysis to show that the separation power of EQGNN can surpass that of GNNs in terms of representing MILP graphs, as shown in Appendix B in **[our current updated PDF](https://openreview.net/pdf?id=KbvKjpqYQR)**. The separation power of GNN is measured by whether it can distinguish two non-isomorphic graphs [2], and it has been shown that GNN has the same separation power as the WL test [3]. Therefore, we first demonstrate why 1-WL test and GNNs fail to distinguish between some non-isomorphic graphs (i.e., components of the foldable dataset), and then demonstrate how our EQGNN distinguishes between them, and thus show that EGNN can surpass the separation power of GNNs.
>
> We present a variant of the WL test specially modified for MILP that follows the same lines as in (Chen et al. 2023b, Algorithm 1) [2].
>
> **Algorithm 2:** WL test for MILP-Graphs
>
> **Input:** A graph instance $(G, H) \in \mathcal{G}_{q,p} \times \mathcal{H}^V_q \times \mathcal{H}^S_p$, and iteration limit $L > 0$.
> 1. Initialize with $c_v^0 = hash(\boldsymbol f^V)$ for all $v\in V$, $c_s^0 = hash(\boldsymbol f^S)$ for all $s\in S$.
> 2. **for** $l = 1,2,...L$ **do**
> 3. $\quad c_{v_i}^l = hash(c_{v_i}^{l-1}, \sum_{j=0}^{p-1} A_{i,j} hash(c_{s_j}^{l-1}))$, for all $v \in V$
> 4. $\quad c_{s_j}^l = hash(c_{s_j}^{l-1}, \sum_{i=0}^{q-1} A_{i,j} hash(c_{v_i}^{l-1}))$, for all $s \in S$
> 4. **end for**
> 5. **return** the multisets containing all colors $\\{\\{c_v^l: v\in V, c_s^l: s\in S \\}\\}$.
>
> (to be continued)

---

> ### Author Response · Authors · 2023-11-19
> **Response to Reviewer jrPx (Part 2)**
>
> The MILP graph is represented by $G = (V \cup S, A)$ and
> $\mathcal{G}_{q,p}$
> as the collection of all such weighted bipartite graphs whose two vertex groups have size $q$ and $p$, respectively. All the vertex features are stacked together as
> $H = (\boldsymbol{f}^V_1,...,\boldsymbol{f}^V_q,\boldsymbol{f}^S_1,...,\boldsymbol{f}^S_p) \in \mathcal{H}^V_q \times \mathcal{H}^S_p$.
> Thereby, the weighted bipartite graph with vertex features
>
> $(G, H) \in \mathcal{G}_{q,p} \times \mathcal{H}^V_q \times \mathcal{H}^S_p$
>
> contains all information in the MILP problem. $hash(·)$ is a function that maps its input feature to a color in $\mathcal C$. The algorithm flow can be seen as follows. First, all nodes in $V \cup S$ are assigned to an initial color $c_v^0$ and $c_s^0$ according to their node features. Then, for each $v_i \in V$, $hash$ function maps the previous color
> $c_{v_i}^{l-1}$
> and aggregates the color of the neighbors of $\\{c_{s_j}^{l-1}\\}_{{s_j}\in \mathcal N(v_i)}$.
>
> Similarly, $hash$ function maps the previous color $c_{s_j}^{l-1}$ and aggregates the color of the neighbors of $\\{c_{v_i}^{l-1}\\}_{{v_i}\in \mathcal N(s_j)}$. This process is repeated until $L$ reaches the maximum iteration number. Finally, a histogram $h_G$ of the node colors can be obtained according to $\\{\\{c_v^l: v\in V, c_s^l: s\in S \\}\\}$, which can be used as a canonical graph representation. The notation $\\{\\{·\\}\\}$ denotes a multiset, which is a generalization of the concept of a set in which elements may appear multiple times: an unordered sequence of elements. That is, the 1-WL test transforms a graph into a canonical representation, and then if the canonical representation of two graphs is equal, the 1-WL test will consider them isomorphic.
>
> Next, we show a case where the 1-WL test fails. Taking [Figure 2](https://postimg.cc/R64RKWvr) in our paper as an example, we feed (a) and (b) from Figure 2 into this algorithm separately and compare the differences at each step. In the setting of Figure 2,
> $\{\boldsymbol{f}^V_i\}$ are equal, $\{\boldsymbol{f}^S_j\}$ are equal, are equal, and the weights of all edges are equal. In these two graphs, each node has two neighbors, but the only difference lies in their topological structure. Initially, each variable vertex has the same color $c_v^0$, and each constraint vertex has the same color $c_s^0$, because their features are the same. This step is the same for Figure 2(a) and Figure 2 (b). Then, each vertex iteratively updates its color based on its own color and information from its neighbors. Note that, since each $v_i$ is connected to two $s_{j_1}$, $s_{j_2}$ in both Figure 2(a) and 2(b), they will also obtain the consistent $c_v^l$ and $c_s^l$ at this step. That is, Figure 2(a) and 2(b) will get the same representation by the algorithm, and they cannot be distinguished by 1-WL test GNNs. Moreover, Xu et al. [3] show that **GNNs are *at most* as powerful as the 1-WL test** in distinguishing graph structures. Therefore, a pair of 1-WL indistinguishable graphs are indistinguishable by GNNs.
>
> For EQGNN, the subfigure (a) and (b) of Figure 2 can be clearly distinguished. Since the two graphs differ only in the connectivity of edges, we only consider the modules associated with edges in EQGNN, i.e., $U_{g1}$ and $U_{g2}$. Specifically, $U_{g_1}(G,\beta) =  \text{exp}(-\textbf{i}(\sum_{(i,j)\in \mathcal E} (A_{i,j}+\beta) Z_{2i}Z_{2q+j}))$, where $i\in\\{0,1,...,q\\}$ indicates the index of variable nodes and $j\in\\{0,1,...,p\\}$ indicates the index of constraint nodes. Since two qubits represent one variable node, $2q$ qubits are used to represent all variables, and the index of the qubit representing the constraint node is numbered from $2q$. In subfigure (a) node $v_3$ is connected to $s_3$ and $s_4$, while in subfigure (b) node $v_3$ is connected to $s_3$ and $s_1$. Therefore, the difference between $U_{g_1}(G_a,\beta)$ and  $U_{g_1}(G_b,\beta)$ includes
> $\text{exp}(-\textbf{i}((A_{3,4}+\beta) Z_{6}Z_{16}))$
> and $\text{exp}(-\textbf{i}((A_{3,1}+\beta) Z_{6}Z_{13}))$.
> Since the edge weights and the parameter $\beta$ are equal, we can just compare $\text{exp}(-\textbf{i}(Z_{6}Z_{13}))$ and $\text{exp}(-\textbf{i}(Z_{6}Z_{16}))$. These two values are obviously not equal, so $U_{g_1}(G_a,\beta) \neq U_{g_1}(G_b,\beta)$ and the overall model $U(G_a,H) \neq U(G_b,H)$. Thereby, EQGNN can distinguish such graphs that the 1-WL test and GNNs cannot.
>
> We have added the above to the revised paper, see Appendix B.
>
> References:
>
> [3] Keyulu Xu, Weihua Hu, Jure Leskovec, and Stefanie Jegelka. How powerful are graph neural networks? In International Conference on Learning Representations, 2019

---

> ### Author Response · Authors · 2023-11-19
> **Response to Reviewer jrPx (Part 3)**
>
> >***Weakness 3: In Section 4.1, it is unclear why and how the number of parameters are compared between neural networks in classic and quantum computers. The authors need to clarify how the criteria are chosen. In Section 4.2, it is also unclear why neural networks in classical computers with 2,096 and 7,904 parameters are chosen as the baseline. It will be better if the authors can compare the proposed model with a non-data-driven method for solving MILP.***
>
> **A3:** Thanks for your suggestion. The GNN of (Chen et al., 2023b) has one hyperparameter that controls the number of parameters, i.e., the embedding size $d$. In the tasks of predicting feasibility, the number of their parameters is $30d^2 +30d$. Our model also has a hyperparameter to control the number of parameters, i.e., the number of blocks $T$. Therefore, in our experiments, we vary these two hyperparameters separately to compare their performance results. In section 4.1, we choose $d = 4, 8, 16, 32$ as the embedding size of GNN, and $T= 2, 4, 6, 8$ as the number of blocks of EQGNN. In our experiments, we first gradually increase the embedding_size/blocks to test the performance of the models and then find the embedding_size/blocks $d^*$ or $T^*$ corresponding to the best performance. Then, we select the values near $d^*$ or $T^*$ and show their corresponding results in the experiments of the paper. In Section 4.2,  for the sake of clarity in the curve graph, we only select two hyperparameters of each model for comparison. We trained EQGNN with the number of blocks at $6$, $10$, and the number of parameters is $88$ and $148$. The embedding size $d$ of GNN is chosen as $8$ and $16$ with the number of parameters at $2$,$096$ and $7$,$904$. The reasons for the setting are as follows. We observe that the train performance of GNN increases as the number of parameters increases, but the generalization performance decreases. The train performance of GNN with $d=8$ is worse than that of EQGNN, and the train performance of GNN with $d=16$ is better than that of EQGNN. Therefore, we choose the GNNs with these two hyperparameters for comparison. We have added the above description in the latest PDF. Furthermore, we have added more numerical experiments to show the performance change of the GNN with different parameters in the Appendix.
>
> In addition, current non-data-driven methods for solving MILP, such as Gurobi or SCIP solver, can use brute-force search to find the exact solution of our datasets, because the size of our data set is relatively small. So we didn't compare them in the experiment.
>
> >***Weakness 4: Throughout the paper, there are many typos/errors/inconsistencies. For example, at the beginning of the third paragraph of Section 2: "Foladable MILP instances"; in equation (4) the authors use $R_x$ gate for encoding $l$ and $\epsilon$ but in Figure 5 $R_z$ is used to encode  $l$ and $\epsilon$ . It would be better to have consistent annotations for example in Section 2, the same $q$ decision variables and constraints $p$ can be used to index $V$ and $S$ in bipartite graph representation. There are many of these problems.***
>
> **A4:** Thanks for pointing out our mistakes. "Foladable MILP instances" should be corrected to "Foldable MILP instances", and  Figure 5 should be corrected to [this picture](https://postimg.cc/5X9jtV7f).
>
> We have rewritten the equation (4) in our latest submitted PDF. In addition, we have modified the vertex set of bipartite graph representation as $V \cup S$, where $V=\\{v_0,\cdots,v_i,\cdots, v_{q-1}\\}$ with $v_i$ representing the $i$-th variable and $S=\\{s_0,\cdots, s_j, \cdots, s_{p-1}\\}$ with $s_j$ representing the $j$-th constraint, and $q$ and $p$ represent the number of decision variables and constraints, respectively. We apologize for not checking these errors carefully, and have checked the full text for such problems, and highlighted them in the latest submitted PDF.
>
> >***Question 3: The author should clearly describe how the VQC learning problems are formulated for MILP feasibility/optimal solutions/optimal values based on the proposed EQGNN respectively either in the main text or appendix. The author should also clearly specify how the binary measurements can be used to approximate the optimal value which is often a non-discrete value. If multiple measurements are performed to recover the approximated quantum state of the VQC, the author should also clearly specify how many measurements are needed for the results provided in the experiment sections.***
>
> **A5**: Thank you for your suggestion. Next, we provide a detailed explanation and add it to our revised paper. First of all, it needs to be noted that our final measurement phase uses Pauli-Z measurement on the single qubit. That is, *there is no need to* recover the output quantum state of the entire quantum circuit by techniques like quantum state tomography, which may require an exponentially large number of measurements.
>
> (to be continued)

---

> ### Author Response · Authors · 2023-11-19
> **Response to Reviewer jrPx (Part 4)**
>
> In our model, we set the number of features encoded per qubit as $\omega$. If $\omega = 2$, each variable has four features, so it is represented by two qubits, as the node $v_0$ in Figure 5. Each constraint has two features, so it is represented by one qubit. Then, at the end of the circuit, we only measure the first qubit representing each node. The measurements corresponding to Figure 5 are shown at the end of the overall architecture diagram in Figure 3. As we can see, the measurement operation of the model acts on $q+p$ qubits, and we can obtain $q+p$ output values, where $q$ and $p$ are the numbers of decision variable nodes and constraint nodes, respectively.  The proposed model can be described as a mapping
> $\Phi: \mathcal{G}_{q,p} \times \mathcal{H}^V_q \times \mathcal{H}^S_p \rightarrow \mathbb{R}^{q+p}$,
>
> i.e., $\Phi(G,H) = \\{\langle 0|U_{\boldsymbol \theta}^{\dagger}(G,H)|O_i|U_{\boldsymbol \theta}(G,H)|0\rangle\\}_{i=0}^{q+p-1}$,
>
> where ${\boldsymbol \theta}$ denotes the set of trainable parameters $(\boldsymbol \alpha, \boldsymbol \beta, \boldsymbol \gamma)$, and $U_{\boldsymbol \theta}(G,H)$ is the unitary matrix of the proposed model. $O_i$ represents $i$-th measurement, e.g., when $\omega$ is equal to 2, $O_0 = Z_0 \otimes I_1 \otimes ... \otimes I_{2q+p-1}$ indicates that Pauli-Z measurement is acted on the first qubit representing the *first* variable, and  $O_1 = I_0 \otimes I_1 \otimes Z_2 \otimes... \otimes I_{2q+p-1}$ indicates that Pauli-Z measurement is acted on the first qubit representing the *second* variable, and $O_{q+p-1} = I_0 \otimes I_1 \otimes... \otimes Z_{2q+p-1}$ for the last constraint node. The output of EQGNN is defined as
> $\\{{\Phi_i(G,H)}\\}_{i=0}^{q+p-1}$.
>
> For predicting feasibility, optimal value and optimal solution, we defined
> $\\phi_{sol}(G,H) = \\{{\\Phi_i(G,H)}\\}^{q-1}_{i=0}$,
>
> and $\phi_\text{{fea}}(G,H) = \phi_\text{{obj}}(G,H) =  \sum_{i=0}^{q+p-1} {\Phi_i(G,H)}$.
> As we can see,  the three tasks use the same structure of EQGNN and the same measurements, but use different ways to utilize the information obtained by measurements.
>
> For details on their loss function and evaluation metric, please see our highlights in Section 3.5 of [our latest uploaded PDF](https://openreview.net/pdf?id=KbvKjpqYQR), because some formulas are unable to show up in this rebuttal box.
>
> >***Question 1:  Is that possible for two MILP problems with different feasible/optimal solutions to be encoded into the same quantum state with the same VQC parameters?***
>
> **A6**: Thanks for your question. First of all, it is known that if the feasible/optimal solutions of two MILP problems are different, their MILP-induced graphs are also not the same. Then, we can answer your question by demonstrating that the proposed EQGNN can encode the two different graphs to different representations. Suppose there are two graphs $\mathcal G_1 = (G_1,H_1) =(V_1 \cup S_1, A_1, H_1)$ and $\mathcal G_2 = (G_2,H_2) = (V_2 \cup S_2, A_2, H_2)$, and the differences between the two graphs may appear in 3 places, i.e., the features of nodes, the weight and connectivity of edges. As we mentioned in Definition 4 in the latest submitted PDF, *there are two types of layers in $U_{\boldsymbol \theta}(G, H)$, one is independent of the input graph, and the other is related to the input graph.* In the proposed EQGNN, the feature encoding layer is $U_x^t(G,H,\alpha_t) = U_{x_1}(c,u,b) \cdot  U_{x_2}(\alpha_t) \cdot U_{x_3}(l,\epsilon,\circ) \cdot U_{x_4}(\alpha_t)$, and the message interaction layer is $U^t_{g}(G,\beta) = U_{g_1}(G,\beta_{t,1}) \cdot U_{g_2}(G,\beta_{t,2}) \cdot U_{g_3} (\beta_{t,3}) \cdot U_{g_1}(G,\beta_{t,4}) \cdot U_{g_2}(G,\beta_{t,5}) \cdot U_{g_4} (\beta_{t,6})$. Among them, only $U_{x_1}(c,u,b)$ and $U_{x_3}(l,\epsilon,\circ)$ are related to the features of nodes, i.e., $H$. And $U_{g_1}(G,\beta_{t})$ and $U_{g_2}(G,\beta_{t})$ are related to the weight and connectivity of edges, i.e., $A$.
>
> * For $A_1 = A_2, H_1 \neq H_2$, the only difference between two quantum circuits is $U_{x_1}(c,u,b)$ and $U_{x_2}(l,\epsilon,\circ)$. Taking $U_{x_1}(c,u,b)$ for example, $U_{x_1}(c,u,b) = \text{exp}(-\textbf{i}(\sum_{i=0}^{q-1}(c_i X_{2i}+u_i X_{2i+1}) + \sum_{j=0}^{p-1} b_j X_{2q+j}))$. If any feature changes, $U_{x_1}$ will change, causing $U_{\theta}(G_1,H_1)$ and $U_{\theta}(G_2,H_2)$ to be different.
>
> * For $A_1 \neq A_2, H_1 = H_2$, there are two cases where the connections of the edges are different or the weights of the edges are different. We have already demonstrated the former in [A2: Response to Weakness 2](https://openreview.net/forum?id=KbvKjpqYQR&noteId=dSaFrTi4MX), and now we focus on the latter. $U_{g_1}(G,\beta) =  \text{exp}(-\textbf{i}(\sum_{(i,j)\in \mathcal E} (A_{i,j}+\beta) Z_{2i}Z_{2q+j}))$. The change of $A_{i,j}$ will lead the change of $U_{g_1}(G,\beta)$, causing $U_{\theta}(G_1,H_1)$ and $U_{\theta}(G_2,H_2)$ to be different.

---

> ### Author Response · Authors · 2023-11-19
> **Response to Reviewer jrPx (Part 5)**
>
> We control the changes in $\mathcal g_1$ and $\mathcal g_2$ individually to demonstrate the difference to our model by that change, and thus to demonstrate that different graphs correspond to different unitary of quantum circuits. We have added the above to our latest uploaded PDF, and please see our highlights in Appendix C.
>
> >***Question 2:  Node permutation equivariance or invariance comes naturally with graph representations. The authors need to clearly state whether the claimed generalizability, expressiveness, and efficiency should be attributed to EQGNN or VQC? What are the main technical contributions? How shall they be positioned in the context of Table 1? Also, why the mentioned reference Schatzki et al. (2022) was not included in Table 1?***
>
> **A7**: On your first question, I may not have fully understood what you meant. Firstly, it needs to be stated that not every VQC is permutation-equivariant. [Figure R1](https://postimg.cc/qhCCzp3P) is a (Hardware efficient ansatz) HEA, which is an intuitive example of failure to satisfy permutation equivariance. As we can see, the output will not be the same when the order of nodes changes. In contrast, we prove the permutation equivariance of our approach in section 3.6 of the latest uploaded PDF. Moreover, if the model is not equivariant, it will overfit the variable/constraint orders in the training data, directly resulting in a decline in generalization performance and expressiveness. As shown in Table 1 of our previous paper, not every existing quantum graph neural network conforms to equivariance, and we make a comparison with QGCN in experiment section 4.8. The experiment shows that the performance of the model without equivariance will be worse. In Tabel 1, QGNN and EQGC are equivariant, but QGNN does not consider the feature encoding of nodes and edges, and EQGC does not consider the information encoding of edges, so they cannot be directly applied to the MILP task. In other words, the generalization and expressiveness brought by our model on the MILP task is unique.
>
> On the second and third questions, Table 1 shows that our model (1) conforms to permutation equivariance, (2) can encode both node and edge features, and (3) can handle both node-level and graph-level tasks. More specifically, compared to these existing QGNNs, the main technical contributions of our approach are (1) the extension of multi-qubit encoding node features to address the high-dimensional feature coding problem, (2) the addition of an optional auxiliary layer to control model expressiveness, and (3) the association of a shared learnable parameter for each feature, and the use of repeated coding mechanisms. We have incorporated the above description into our revised paper. Thank you for your suggestions for making our article clearer.
>
> On the last questions, the role of Table 1 is to compare existing quantum **graph** neural networks, but Schatzki et al. (2022) are not specialized for graph tasks. As we discuss in Appendix A, their equivariant QNNs can used to learn various problems with permutation symmetries abound, such as molecular systems, condensed matter systems, and distributed quantum sensors (Peruzzo et al., 2014; Guo et al., 2020). Moreover, they also haven't made any specific design for the high-dimensional features and edges of the graph, and they cannot be directly used for MILP tasks.
>
> >***Question 4: How expressive is the proposed model?***
>
> **A8**: Thanks for your question. We use a tool presented in [The Power of Quantum Neural Networks](https://www.nature.com/articles/s43588-021-00084-1) to quantify the expressiveness of our model. The authors introduce the **effective dimension** as a useful indicator of how well a particular model will be able to perform on the dataset. In particular, this algorithm follows 4 main steps:
> 1. Monte Carlo simulation: the forward and backward passes (gradients) of the neural network are computed for each pair of input and weight samples.
> 2. Fisher Matrix Computation: these outputs and gradients are used to compute the Fisher Information Matrix.
> 3. Fisher Matrix Normalization: averaging over all input samples and dividing by the matrix trace.
> 4. Effective Dimension Calculation: according to the formula [in this picture](https://postimg.cc/MM0g2XXb) (it is unable to display in this rebuttal box).
>
> We test the proposed model with two blocks on the MILP dataset and use the random weight to evaluate the normalized effective dimension. As shown in [Figure R2](https://postimg.cc/Wd6CrtPJ), with the increase in the number of data, the normalized effective dimension of the proposed model can converge to near 0.75, which shows that our model can achieve good expressiveness even with two blocks. We have added the above to the Appendix of our latest PDF.
>
> We hope that the above answers can address your concern. If you have any further questions, please contact me again. I am looking forward to your reply.

---

> ### Author Response · Authors · 2023-11-22
> **(To Reviewer jrPx) We would love to hear your feedback on our rebuttal**
>
> Dear Reviewer jrPx,
>
> Thanks again for your review. As the discussion period is close to the end and we have not yet heard back from you, we wanted to reach out to see if our rebuttal response has addressed your concerns.
>
> We are more than happy to discuss further if you have any further concerns and issues, please kindly let us know your feedback. Thank you for your time and attention to this matter. We look forward to hearing back from you soon.
>
> Best regards

---

> > ### Comment · Reviewer_jrPx · 2023-11-22
> >
> > I would like to thank the authors for their effort and explanations. The authors have addressed several concerns. However, there are still a few left, or raised in the rebuttal phase:
> >
> > 1. Although the authors removed the previous claim "We both theoretically and empirically show the separation power of EQGNN can surpass that of GNNs in terms of representing MILP graphs." They claim that "EQGNN can distinguish graphs that cannot be distinguished by GNNs, such that it is capable of representing general MILP instances" in their updated paper, with only one exemplar graph which is indistinguishable for traditional GNN but distinguishable for QNN. It would be great if the authors can be clear on any theoretical guarantee and provide more rigorous discussions to support their claims on the quantum advantages.
> >
> > 2. In Appendix C and Response A6, theorem 4, the authors did not provide any discussion on the case when $A_1 \not = A_2, H_1 \not= H_2$.
> >
> > 3. There are a few more typos in the updated paper. For example, on page 5, the last sentence, "loos" --> "loss".

---

> ### Author Response · Authors · 2023-11-23
> **Second Response to Reviewer jrPx**
>
> Dear Reviewer jrPx,
>
> Thanks for your reply. We provide a more rigorous discussion to support our claim as follows. We first analyze why 1-WL or GNNs cannot distinguish two graphs. Then, we improve Theorem 4 (i.e., add a discussion on the case when $A_1 \neq A_2, H_1 \neq H_2$), and use Theorem 4 to show why the proposed EQGNN does not have this problem.
>
> In [A2: Response to Weakness 2 you mentioned](https://openreview.net/forum?id=KbvKjpqYQR&noteId=dSaFrTi4MX) or Algorithm 1 of our paper,  we provided a WL test algorithm for MILP-Graphs, which is consistent with (Chen et al. ICLR 2023) [2]. As we can see, the final output of the algorithm is a histogram $h_G$ of the node colors, which can be obtained by counting the number of different colors $c\in \mathcal C$ in the $\\{\\{c_v^l: v\in V, c_s^l: s\in S \\}\\}$. The notation $\\{\\{·\\}\\}$ denotes a multiset, which is a generalization of the concept of a set in which elements may appear multiple times: an unordered sequence of elements. The histogram $h_G$ is used as the canonical graph representation. The algorithm will consider two graphs with identical histograms are the same. However, the essential issue of the algorithm is that any two nodes may be assigned the same color $c_v$ or $c_s$, which causes two identical histograms may correspond to different two MILPs. Therefore, the key to distinguishing two 1-WL indistinguishable graphs is that the proposed methods can encode the two different graphs to different graph representations, as the goal of our Theorem 4. As we can see, we can associate Theorem 4 to demonstrate our claim. To end this, we further improve Theorem 4, as you mentioned that the case when $A_1 \neq A_2, H_1 \neq H_2$ is not discussed.
>
> The discussion on $A_1 \neq A_2, H_1 \neq H_2$ as follows. As we can see, in the feature encoding layer, we use the $\text{exp}(-\text{i} f (X))$ gate to encode features, and the graph information interaction uses $\text{exp}(-\text{i} b (Z_iZ_j))$, where $f \in H$ denotes features and $b \in A$ denotes edge weights. We specially use different Pauli gates ($X$ and $Z$) as the bases of the two layers, so as to ensure that as long as one of the layers changes, the whole unitary changes.
>
> In addition, as we have mentioned many times, it is well-known that GNNs are at most as powerful as the 1-WL test in distinguishing graph structures [1,2] since they have a similar neighborhood-aggregation mechanism. Therefore, GNNs cannot distinguish 1-WL indistinguishable graphs. Moreover, as mentioned in [2], around 1/4 of the problems in MIPLIB 2017 (Gleixner et al., 2021)[3], the number of colors generated by the WL test is smaller than the number of vertices. Practitioners using traditional GNNs cannot benefit from that if there are these MILPs in their set of interests. Thereby, by Theorem 4, we can demonstrate that EQGNN can distinguish MILP graphs that GNNs cannot distinguish. We have added the above to our updated paper. Thanks for your suggestion!
>
> It's also worth mentioning that we have empirically demonstrated our claim, i.e., Figure 6 shows the results of EQGNN on 1000 pairs of MILPs that cannot be distinguished by GNNs, and the errors of GNN stay 0.5 while that of EQGNN is close to 0. This dataset in this experiment comes from the (Chen et al. ICLR 2023) [2].
>
> Thanks again for your review. May you have a blessed and happy Thanksgiving!
>
> With gratitude
>
> References:
>
> [1] Keyulu Xu, Weihua Hu, Jure Leskovec, and Stefanie Jegelka. How powerful are graph neural networks? In International Conference on Learning Representations, 2019.
>
> [2] Ziang Chen, Jialin Liu, Xinshang Wang, and Wotao Yin. On representing mixed-integer linear programs by graph neural networks. In Proceedings of the International Conference on Learning Representations, 2023
>
> [3] Ambros Gleixner, Gregor Hendel,et.al., MIPLIB 2017: Data-Driven Compilation of the 6th Mixed-Integer Programming Library. Mathematical Programming Computation, 2021

---

### Official Review · Reviewer_pyPd · 2023-10-31

**Soundness:** 3 good
**Presentation:** 3 good
**Contribution:** 3 good
**Rating:** 8
**Confidence:** 4

**Summary:**

This paper proposes a quantum counterpart of GNN, called equivariant quantum GNN (EQGNN), that is tailor-made for solving mixed-integer linear programming (MILP). The key feature of EQGNN is the preservation of the permutation equivariance in the GNN. This feature allows EQGNN to demonstrate better expressiveness compared to other GNNs, in the sense that EQGNN can distinguish pairs of *foldable* MILPs that existing GNN design is not able to. Therefore, EQGNN can accurately predict the feasibility of general MILPs. Extensive numerical experiment results are presented to show that EQGNN has faster convergence and attains better generalization with less data compared to GNNs.

**Strengths:**

This work presents a novel variant of Graph Neural Network (GNN) based on quantum parametric circuits. This so-called Equivariance Quantum Graph Neural Network (EQGNN) consists of the feature encoding layer, graph message interaction layer, and optional auxiliary layer, all expressed as parametrized quantum circuits. This new design allows EQGNN to overcome a fundamental limitation of traditional GNNs (i.e., GNNs can not distinguish pairs of foldable MILP instances). Compared to other quantum GNN architecture, EQGNN incorporates the feature of the edges, which renders it a problem-inspired model and does not suffer from the barren plateau issue.

The numerical results appear to be very strong. Compared to prior arts, EQGNN demonstrates much better separation power for foldable MILP instances. For general MILP tasks, EQGNN has better predictions for the optimal value with much fewer parameters (~100), while traditional GNN requires approx. $\sim 10^4$ parameters to achieve similar performance.

The paper is well-written and the mathematical formulation is easy to follow.

**Weaknesses:**

In section 3.6, the authors claim that "we can prove that the whole model conforms to permutation equivariance by ensuring that each layer conforms to equivariance". However, I was not able to find a theorem statement in the PDF (including appendices).

Many plots use the "rate of errors" as a performance metric. How is this "rate of errors" defined and evaluated in the experiments? Does it require the ground truth of the tested MILP instances? How to get the ground truth?

**Questions:**

1. From the numerical experiments, it appears that an EQGNN deployed on a small- to intermediate-size parametric quantum circuit outperforms a traditional GNN-based model with many more parameters. Can we directly employ this new GNN design on a classical computer to outperform other models requiring similar classical computing resources?

2. Is the equivariance feature unique to quantum parametric circuits? If not, I wonder if it's possible to obtain a quantum-inspired equivariance GNN for MILPs that is native to classical computing architecture but still outperforms traditional GNN models.

---

> ### Author Response · Authors · 2023-11-19
> **Response to Reviewer pyPd**
>
> >***Weakness 1: In section 3.6, the authors claim that "we can prove that the whole model conforms to permutation equivariance by ensuring that each layer conforms to equivariance". However, I was not able to find a theorem statement in the PDF (including appendices).***
>
> **A1:** Thanks for your precious feedback. We have rewritten Section 3.5 and 3.6 and added formulas to Section 3.2 and 3.3. We formulate the proposed model and present four definitions and three theorem statements to prove the permutation equivariance of our proposed model. Since this part is about two and a half pages long, we will not show it in this reply box, please refer to the highlighted section of [our latest uploaded PDF](https://openreview.net/pdf?id=KbvKjpqYQR).
>
> >***Weakness 2: Many plots use the "rate of errors" as a performance metric. How is this "rate of errors" defined and evaluated in the experiments? Does it require the ground truth of the tested MILP instances? How to get the ground truth?***
>
> **A2:** "Rate of errors" is the performance metric of predicting feasibility, and it requires the ground truth of the tested MILP instances. Each MILP instance has a label $y_{fea}$ to represent its feasibility, and $y_{fea}=0$ indicates that the MILP is infeasible, and $y_{fea}=1$ indicates the MILP is feasible. Suppose that the predicted result of our model is $\hat{y}_{fea}$, and we set an indicator function
>
> $\mathbb I_{\hat{y}_{fea}>1/2}$
>
> to represent the predicted feasibility. Then,
> $\text{Rate of errors} = \frac{1}{M}(\sum_{m=0}^{M-1} y^m_{fea} · \mathbb I^m_{\hat{y}_{fea}>1/2}),$
> where $M$ indicates the number of tested MILP instances, and the metric represents the number of correct predictions for feasibility. In addition, the ground-truth is solved by PySCIPOpt, which is a Python interface to the SCIP optimization software. We apologize for not writing this part clearly, and we have added the above description to our latest submitted PDF. Thanks for your suggestion.
>
> >***Question 1: From the numerical experiments, it appears that an EQGNN deployed on a small- to intermediate-size parametric quantum circuit outperforms a traditional GNN-based model with many more parameters. Can we directly employ this new GNN design on a classical computer to outperform other models requiring similar classical computing resources?***
>
> **A3**: We can use TorchQuantum (a Pytorch-based library) to simulate the designed EQGNN on classical computers. Take predicting feasibility of unfoldable dataset as an example, we test our model EQGNN and GNN on the same machine (4 physical CPUs with 224 cores Intel(R) Xeon(R) Platinum 8276 CPU @ 2.20GHz, and a GPU (NVIDIA A100)). We adjust the hyperparameters of EQGNN and GNN separately to achieve their best performance, and then compare their computing resource usage.
>
> |                  | EQGNN (blocks=8) | GNN(embeddig size=6) |
> | ---------------- | ---------------- | -------------------- |
> | GPU Memory Usage | 1219 MiB         | 695 MiB              |
> | Average Runtime  | 0.2699 s          | 0.0311 s               |
> | Rate of error (test set) |  0.0765      |    0.0888       |
>
> As we can see, EQGNN requires 2x GPU Memory Usage and nearly 9x time to outperform GNN, because the simulating quantum algorithms usually involves more complex matrix operations. It is well known that quantum algorithms still cannot be efficiently simulated by classical computers, which is also seen as a quantum advantage. On the other hand, the GNN using more computing resources will not have better results than GNN (embedding size=6).
>
> >***Question 2: Is the equivariance feature unique to quantum parametric circuits? If not, I wonder if it's possible to obtain a quantum-inspired equivariance GNN for MILPs that is native to classical computing architecture but still outperforms traditional GNN models.***
>
> **A4**: Thanks for your question. The equivariance feature is not unique to quantum parametric circuits, and it requires careful design, as shown in section 3.6 of our revised paper. Moreover, we believe it is a promising research direction to design a quantum-inspired equivariance GNN that can outperform traditional GNN models.

---

> > ### Comment · Reviewer_pyPd · 2023-11-22
> >
> > Thanks for the authors' response regarding my comments. The revised PDF includes more theoretical details. I'll keep my score.

---

### Author Response · Authors · 2023-11-21
**General Response by Authors**

Dear area chair and reviewers,

We thank all reviewers for their valuable reviews that have helped improve our submission. We especially thank reviews for recognizing that our paper can
1) **overcome a fundamental limitation** of traditional GNNs (Reviewer  pyPd),
2) **carefully**  chosen gates for permutation equivariance (Reviewer  jrPx),
3) solve mixed-integer LP in an **interesting and unique** way (Reviewer  AMEi),
4) study the **essential** trainability of their model (Reviewer 43Zw).

However, the main concerns include the separation power of EQGNN (jrPx), the setting of parameter counts of GNN (jrPx, 43Zw), and the motivation of ansatz (AMEi). We briefly introduce the responses to these concerns in this general response section and provide concrete mathematical formulation, proofs, and numerical experiments in the part of the response to each reviewer.

**For the separation power of EQGNN**,  we have added theoretical analysis about why EQGNN can distinguish two graphs that cannot be distinguished by GNNs and have provided a statement that these GNN indistinguishable graphs are common in the general MILP datasets. Based on this, we can show the quantum advantage and the importance of the proposed model for solving MILP tasks.

**For the setting of parameter counts of GNN**, we have added an explanation that the parameter counts of the GNN depend on the embedding size $d$, and we choose a few embedding sizes around the embedding size that makes the test performance of GNN the best for presentation. In addition, we added numerical experiments to show that 1) the training and testing performance of GNN is worse than that of EQGNN even with a similar number of parameters; 2) the generalization error of GNN increases with the parameter counts, so that it cannot achieve better test performance than EQGNN. Note that the above experiments are conducted on unfoldable MILP datasets, which contain all MILP graphs that can be distinguished by GNN. In other words,  EQGNN still has an advantage for MILP graphs that can be distinguished by GNN.

**For the motivation of ansatz**,  we have added four definitions and three theorem statements to prove the permutation equivariance of our proposed model, and have shown the motivation of the multi-qubit encoding mechanism, parameter sharing mechanism, and the choice of parametric gates for learning graph information interaction.

We hope that our responses can address the concerns of all reviewers and that reviewers can have a more comprehensive understanding of our paper.  As the discussion period is close to the end we have not yet heard back from reviewers. We are more than happy to discuss further if you have any further concerns and issues, please kindly let us know your feedback. Thank you for your time and attention to this matter. We look forward to hearing back from you soon.

Best regards

---

### Author Response · Authors · 2023-11-21
**We would love to hear your feedback on our rebuttal**

Dear Reviewer,

Thanks again for your review. As the discussion period is close to the end and we have not yet heard back from you, we wanted to reach out to see if our rebuttal response has addressed your concerns.

We are more than happy to discuss further if you have any further concerns and issues, please kindly let us know your feedback. Thank you for your time and attention to this matter. We look forward to hearing back from you soon.

Best regards

---

### Public Comment · ~Kun_Li10 · 2026-01-16
**The code is not able to run.**

ValueError: Number of circuit parameters 25 mismatch with sum of num inputs and weights 26 #1
opened on Nov 26, 2025

`
After creating the virtual environment, running python train.py will raise 'ValueError: Number of circuit parameters 25 mismatch with sum of num inputs and weights 26' in the qgnn.py file. The position of the error is as follows
self.qnn = EstimatorQNN( circuit=self.circuit, input_params=self.encoder.parameters[:] + self.conv_feature.parameters[11:] + self.entanglement[1:], weight_params=self.conv_feature.parameters[:11] + self.entanglement[:1], input_gradients=True, )
Could you please offer a solution to the error message?
`

Me too, what is wrong?

---

### Meta-Review · Area_Chair_6Mdc · 2023-12-05

**Metareview:**

This paper studies mixed-integer linear programming (MILP), an essential task for operation research but is NP-hard in general. This paper studies machine learning-based models for solving MILP, in particular it generalizes an approach by Chen et al. 2023 using GNN to the quantum setting, called equivariant quantum GNN (EQGNN). EQGNN can guarantee to distinguish two MILPs, i.e., leading to different graph embeddings, and it designs a novel quantum parametric circuit that can encode node and edge features while maintaining the property of permutation equivariance. Compared with classical GNN, EQGNN can achieve better separation power and generalization performance with fewer parameters.

The strengths, as pointed out by the reviewers, are:
- The numerical results are appealing. In particular, EQGNN has better predictions for the optimal value with ~100 parameters, while traditional GNN requires ~10^4  parameters to achieve similar performance.
- The idea of exploring problem-inspired ansatzes motivated by the training data structure, i.e., extending the idea of learning to Optimize from classical ML to quantum ML/NISQ algorithms is novel.

On the other hand, the paper also has weaknesses:
- Reviewers found the conclusion that the numerical results demonstrating a separation in expressive power between the EQGNN and the classical GNN by Chen et al. may need more supports. In particular, the comparison to other methods beyond data-driven methods (the classical GNN and the quantum EQGNN) can be more convincing. The authors replied in the rebuttal that current non-data-driven methods for solving MILP, such as Gurobi or SCIP solver, can use brute-force search to find the exact solution of our datasets. Because the size of  data set is relatively small, this is not included in the experiments.
- Reviewers found the ansatz not principled enough. For the feature encoding layer, the use of angle encoding means their encoding does not uniquely map an MILP instance to an initial quantum state. Meanwhile, the permutation invariance which follows from choosing  rotations in the message interaction layer seems to arbitrary and not well-explained overall.

Reviewers also questioned about theoretical analysis in the initial review, but this is much better explained in the version after rebuttal.

During the rebuttal, the AC and reviewers discussed about the overall contribution by this work, as the score is on the borderline. The conclusion is to reject this paper. A main reason is that compared to other papers on the borderline, this paper is not very first-principled in terms of contributions in machine learning: The Chen et al. (2023)'s paper was a ICLR work because it applies a learning idea, namely GNN, to solving MILP. But for this paper, the contribution is more on the quantum side (a quantum version of GNN) and operations research side (solve MILP better, and can solve foldable MILP). The concern is especially notable in the sense that MILP is not a learning problem per se, and the authors did not make detailed comparisons to non-data driven approaches. The contribution to machine learning seems to be not significant enough to meet the bar of ICLR. In addition, there are also the above weaknesses pointed out by the reviewers.

A typo in abstract: mixted -> mixed

**Justification For Why Not Higher Score:**

See my explain above for the weaknesses and my reason of rejection.

**Justification For Why Not Lower Score:**

N/A

---

### Decision · Program_Chairs · 2024-01-16

Reject